# Novel metrics reveal new structure and unappreciated heterogeneity in *Caenorhabditis elegans* development

**Gunalan Natesan[1], Timothy Hamilton[2], Eric J. Deeds[2,3]\*, Pavak K. Shah[1,3]\***

**1** Department of Molecular, Cell and Developmental Biology, University of California, Los Angeles, California, United States of America, **2** Department of Integrative Biology and Physiology, University of California, Los Angeles, California, United States of America, **3** Institute for Quantitative and Computational Biosciences, University of California, Los Angeles, California, United States of America

\* deeds@ucla.edu (EJD); pavak@ucla.edu (PKS)

## Abstract

High throughput experimental approaches are increasingly allowing for the quantitative description of cellular and organismal phenotypes. Distilling these large volumes of complex data into meaningful measures that can drive biological insight remains a central challenge. In the quantitative study of development, for instance, one can resolve phenotypic measures for single cells onto their lineage history, enabling joint consideration of heritable signals and cell fate decisions. Most attempts to analyze this type of data, however, discard much of the information content contained within lineage trees. In this work we introduce a generalized metric, which we term the branch edit distance, that allows us to compare any two embryos based on phenotypic measurements in individual cells. This approach aligns those phenotypic measurements to the underlying lineage tree, providing a flexible and intuitive framework for quantitative comparisons between, for instance, Wild-Type (WT) and mutant developmental programs. We apply this novel metric to data on cell-cycle timing from over 1300 WT and RNAi-treated *Caenorhabditis elegans* embryos. Our new metric revealed surprising heterogeneity within this data set, including subtle batch effects in WT embryos and dramatic variability in RNAi-induced developmental phenotypes, all of which had been missed in previous analyses. Further investigation of these results suggests a novel, quantitative link between pathways that govern cell fate decisions and pathways that pattern cell cycle timing in the early embryo. Our work demonstrates that the branch edit distance we propose, and similar metrics like it, have the potential to revolutionize our quantitative understanding of organismal phenotype.

## Author summary

Lineage tracing has seen a renaissance as imaging and molecular technologies have made it possible to perform increasingly rich quantitative experiments in developing systems. Although the joint capture of cellular phenotypes and lineage history enables us to study how important developmental events are regulated, the volume and complexity of the

**Data Availability Statement:** All novel code needed to replicate our analysis is provided through a Github repository (https://github.com/shahlab-ucla/graph_distances) and archived on

Zenodo, DOI: 10.5281/zenodo.10127577. Lineage data used in this study was retrieved from www.digital-development.org/download.html, the digital data repository provided by Du et al. DOI: 10.1016/j.devcel.2015.07.014.

**Funding:** This work was supported by NIH grant R21DC019485 (PKS). The funders had no role in study design, data collection and analysis, decision to publish, or preparation of the manuscript.

**Competing interests:** The authors have declared that no competing interests exist.

data produced make it difficult to systematically discover new patterns and relationships from this data. We have developed a new way of measuring how cellular phenotypes, such as the length of the cell cycle, differ between cell lineages and applied this approach to the characterization of embryonic development in *Caenorhabditis elegans*, a microscopic roundworm that has long been used as a model system for studying the regulation of cellular differentiation during embryonic development. Our quantitative and unbiased approach allowed us to describe previously unknown patterns of cell cycle timing between the major lineages of the *C. elegans* embryo, discover surprising differences between populations of wild type embryos and between embryos in which a panel of genes essential for embryonic development had been perturbed, and provided a quantitative link between cell fate and cell cycle timing patterns that have been widely observed in development but not well understood. These findings highlight the power of our approach and motivate continued investigation of the links between cell cycle timing and cell fate in developing embryos and stem cells.

## Introduction

The differentiation of cell types in the developing embryo depends on both cell autonomous processes and signaling from neighbors, diffusible cues, and mechanical forces. In metazoa, lineal history plays an important role in patterning many of these factors and thus in establishing the basic animal body plan [1]. The study of cell lineages in eutelic organisms, which possess a fixed number of somatic cells and thus exhibit stereotypical cell lineages, has been a powerful driving force in our understanding of fundamental developmental and biological processes [2,3]. Cell lineages in these organisms represent valuable scientific resources, providing a spatial and temporal index of the animal onto which multimodal measurements can be aligned [4]. Aligning measurements such as gene expression [5,6], chromatin accessibility [7], cell size and shape [8], and the effects of genetic perturbations [9–12] onto the *Caenorhabditis elegans* lineage has contributed to an increasingly holistic view of development. Advances in light microscopy and computer vision have dramatically expanded the reach of these approaches, and datasets are now available containing measurements aligned to thousands of embryonic cell lineages [10,13]. The scale of these data poses interesting challenges for data exploration and analysis.

Cell lineages map intuitively to mathematical graphs, and the alignment of cell divisions along body axes in *C. elegans* allows for unambiguous names to be assigned to each cell produced from a division [3]. This allows *C. elegans* cell lineages, and those of any other eutelic species, to be considered as *ordered binary trees*, a type of graph that allows straightforward one-to-one alignments to be made between any pair of lineages within or between individual embryos. This property dramatically simplifies the application of metrics computed on lineage trees, since a single unique value can be calculated for each comparison. While the inference of lineage relationships is common practice in the study of evolutionary relationships [14,15], the distinct problem in comparing phenotypic measurements aligned to lineages has been less extensively explored as relevant studies of developmental timing use summary statistics based on linear regressions [11,12,16,17]. Comparisons of the topology of cell lineages has been previously performed using the Robinson-Foulds distance and triplet distance, which each rely on the generation and comparison of sub-trees, accumulating a count of shared sub-trees between lineages normalized against the total number of possible sub-trees to arrive at a metric [18].

In this work, we developed several metrics that operate on the topology of lineages and applied them to the analysis of the *C. elegans* embryonic cell lineage. The first of these is the tree edit distance [19], which has previously been applied to the comparison of neuron morphology [20] and RNA secondary structure [21]. The tree edit distance allows one to quantify the topological changes in lineage between different embryos, different sublineages, or different experimental conditions. For ordered binary trees, the tree edit distance can be computed more efficiently and is more directly interpretable than previously used sub-tree-based metrics [19]. The second metric that we developed is the "branch distance," which measures the similarity between lineages based on quantitative measurements of properties of either cells or branches within the lineage. In this case, our particular focus was on the timing of cell division events in the lineage.

Both metrics represent intuitive notions of distance, which greatly aids their use in downstream analyses such as unsupervised clustering and hypothesis testing. We benchmarked these metrics using a published database of wild type and RNAi-perturbed *C. elegans* embryonic cell lineages [10]. We first demonstrate the statistical effects of using cell birth/division timing as a measure of developmental time, motivating the use of cell cycle duration [11,12]. Our analysis also describes previously uncharacterized heterogeneity in wild type lineages and in the phenotypic consequences of RNAi variability on developmental timing. Others have previously demonstrated that sibling asynchrony in division timing reflects the signaling history of that lineage [11]. We apply the branch distance to measure lineage-wide patterns in cell cycle timing and show that Notch signaling is responsible for producing a striking pattern of similarity among the anterior cell lineages of the embryo. Finally, we apply this approach to a systematic analysis of RNAi perturbations that result in cell fate transformations where we find that, while developmental timing appears to be highly sensitive to genetic perturbation, RNAi against genes in a subset of important developmental regulators generate transformations that preserve lineage-specific developmental clocks.

## Results

### Defining metrics on spaces of cell lineages

Multicellular organisms develop from a single cell through a sequence of divisions. The stereotypical nature of *C. elegans* development makes it possible to uniquely identify every cell in its somatic lineage based on the orientation of the division of its predecessor relative to the embryo's body axes [3]. This feature of *C. elegans* development has been a major advantage of its use as a model system, enabling systematic and quantitative studies of developmental processes. Here, we take advantage of the structured nature of the *C. elegans* cell lineage to represent it digitally as an ordered binary tree, such that nodes and edges represent cells and division events respectively. The cells in the embryo and the corresponding nodes in the tree can be labeled using the convention based on the orientation of cell divisions along body axes [3] and can be associated with quantitative measurements on a cell-by-cell basis. This natural representation of the lineage as a binary tree suggests several straightforward metrics for comparing lineages to quantify how, say, a gene knockout impacts development (Fig 1A). The first is the *tree edit distance*, which is derived from the graph edit distance in graph theory and is based on counting the minimum number of operations (such as adding or removing a node or edge) that is needed to convert one tree into another. Since *C. elegans* development is stereotypical, there is a natural alignment between any two trees based on the naming convention described above [16]. This makes computing the tree edit distance very straightforward, essentially reducing the calculation to determining the number of nodes that are different between the two trees (Fig 1B), a notion similar to other measures, such as the Robinson Foulds metric

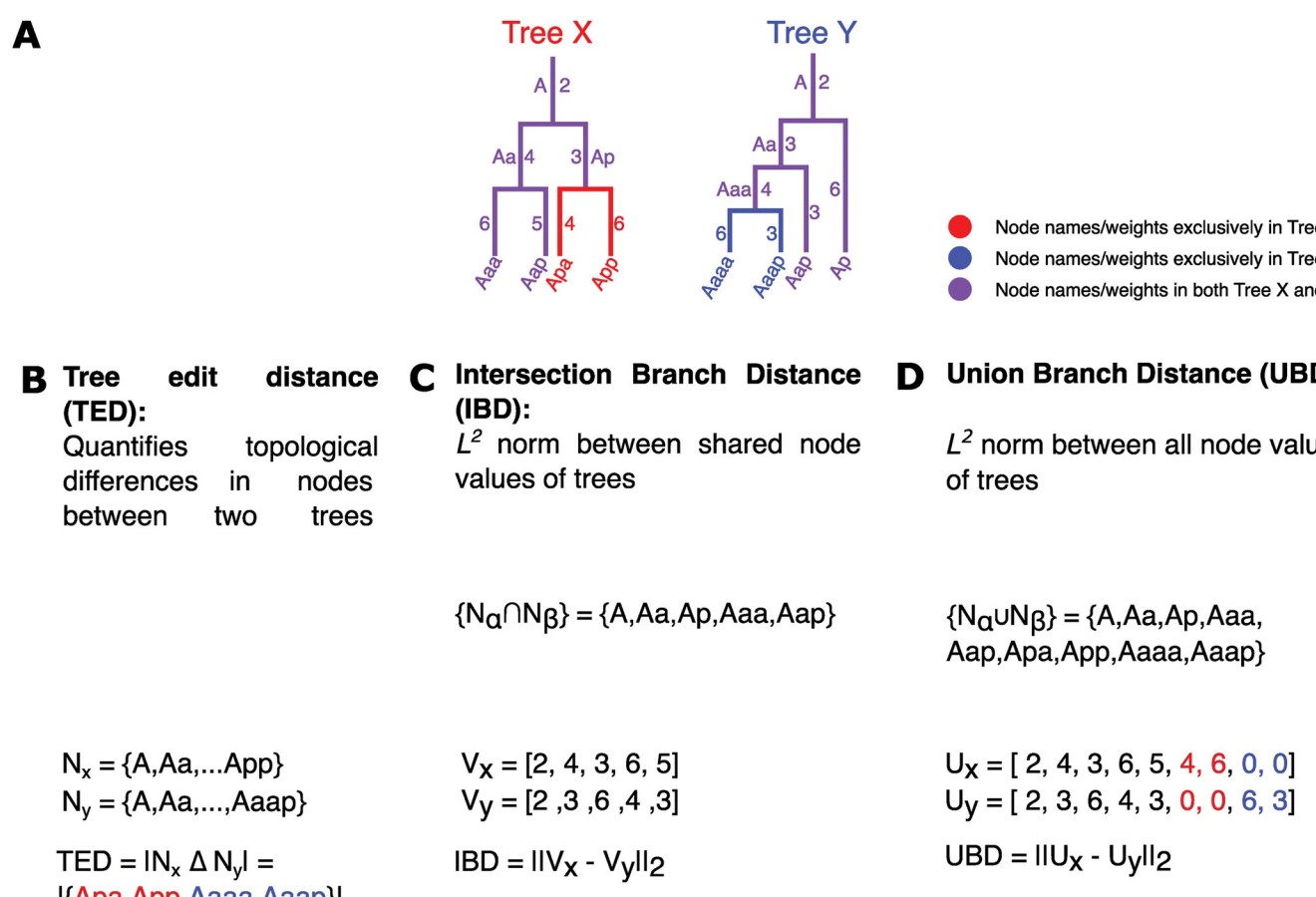

**Fig 1. Defining Distance Metrics on Lineage Based Tree Structures. (A)** Cell lineages can be expressed as binary trees, with parent, sibling, and child cells relationship reflected in node topology. *C. elegans* has a naming convention that allows for direct comparisons between cells in distinct lineages. Here we show schematics of two lineages with different topologies, corresponding to embryos "X" and "Y." The canonical names of the cells are shown either next to or at the end of the corresponding edges. Lineage tracing data also provides information about how long each cell persists between when it is "born" through division and when it divides itself. The numerical values next to each edge indicate these cell cycle times in this schematic. **(B)** The tree edit distance describes the topological differences between trees by counting the number of additive/subtractive operations required to transform one tree into another. In the case of *C. elegans* lineage trees, this corresponds to the size of the symmetric difference between the set of nodes present in one embryo vs. another. **(C)** The intersection branch distance is the Euclidean or $L^2$ norm between measurements associated with shared nodes or edges of trees, disregarding topological differences between trees by only considering nodes/edges present in both trees. **(D)** The union branch distance is the Euclidean or $L^2$ norm between values on the union set of nodes or edges between trees. Nodes or edges that are absent from one tree in any comparison are given a 0 value.

(S1 Appendix). This metric captures how perturbations like gene knockouts influence the *topology* of the lineage.

The second is a class of metrics that can be computed on any measurement associated with individual cells within the tree, such as gene expression or time. Again, here we use the stereotyped nature of *C. elegans* development to directly align any two lineage trees on a cell-by-cell basis, meaning that a single and unique distance can be calculated for any comparison between lineages. Given this alignment, any numerical property of cells during development can be unambiguously converted into a vector representation (Fig 1C). This allows us to use any metric on such vectors to compare the trees. The most straightforward metric that might be used is simply the Euclidean distance (i.e., the $L^2$ norm). Here we focus on the application of the $L^2$ norm to comparisons of cellular division timing between embryos, which has been shown to vary under genetic perturbation [16], and is a measurement produced by every method for lineage tracing by cell tracking. We call this metric the "*branch distance*" since these division

timings represent the length of the branches in the tree (Fig 1C). Of course, a gene knockout or mutation might impact *both* the topology of the tree and the timing of cell divisions. To account for this, we need to have a way for dealing with cases where a cell/node exists in one tree, but not in the other (Fig 1C). Here, we define two different types of branch distance to account for this problem. The first is the *intersection branch distance*, where the vector of division timings is constructed only on the basis of cells that are shared between two lineages (i.e., they are in the *intersection* of the list of cells in the two) (Fig 1D). The second is the *union branch distance*, where we simply set the cell cycle timing of any cell that is missing from any embryo to 0 and calculate the distance in the normal way (Fig 1D). The union branch distance thus captures differences in both timing and topology, as topological differences are reflected in the absence of cells and thus cycle times. Unpaired cell cycle times are squared and added to the branch distance, as the existence of a pair to that cell would only add the square of the difference of those values. This is functionally equivalent to imputing a zero for missing nodes. As described below, these two metrics capture different aspects of variation between trees.

## The branch distance reveals unexpected batch effects in WT embryos

The most straightforward method for cell lineage tracing is via direct observation and cell tracking. Even in the absence of visible reporters, this approach inherently generates both spatial (3D cell positions and cell trajectories) and temporal (the timing of cell divisions) measurements of the embryo. The distribution of individual cell cycle times within lineages have been deeply explored at a single cell resolution [11,12,22], making hierarchically structured lineage data an attractive target for analysis via our branch metrics. During lineage tracing experiments, every cell in developing embryos is tracked, and each cell division can be mapped to a particular time $t$, with $t = 0$ corresponding to, say, the first division of the zygote. There are thus two ways of thinking about the "branch length" value for each cell in the tree (Fig 1C). In one scenario, we could label each cell with its "birth time," which is just the time $t$ at which the cell was generated through a division event. Another alternative is to consider the "cycle time" for that cell, which is just the length of time between when the cell is born through a division event until it divides itself.

  Prior work has claimed that developmental timing in C. elegans is highly coordinated, drawn primarily on comparisons of cell birth times [16,17]. In particular, Bao et al. showed that birth times for cognate cells from different embryos are highly correlated, with $R^2$ values that range between 0.995 and 0.997 (Fig 2A) [16]. While comparing birth times between embryos seems natural, there is a potential issue with that approach. In particular, the birth time of a given cell is the sum of the cell cycle times of all the previous division events (Fig 1C). Since there is some randomness in these cycle times, we can think of those times as random variables, noting that summing over random variables always reduces variation [23]. In other words, the "birth time" is essentially equivalent to averaging the previous cycle times, and averaging generally suppresses variation (i.e., the standard error of the mean is generally less than the standard deviation). In addition, because the birth times are a sum of previous cycle times, birth times for cells born later in development will always be larger than birth times for cells born earlier. Both effects can spuriously increase the correlation in cell birth times between embryos. To demonstrate this, we completely randomized the cell cycle times in the embryo, intentionally destroying any correlation in the length of cell cycles for the same cells across each randomized embryo (see Methods). After randomizing these cycle times, we found birth time correlations with $R^2$ values between 0.65 and 0.85 (Fig 2B), despite a complete absence of correlation in the individual cell cycle times (Fig 2C). Thus, while the cycle times are still highly correlated between WT embryos ($R^2$ between 0.97 and 0.99, Fig 2D), the correlation is

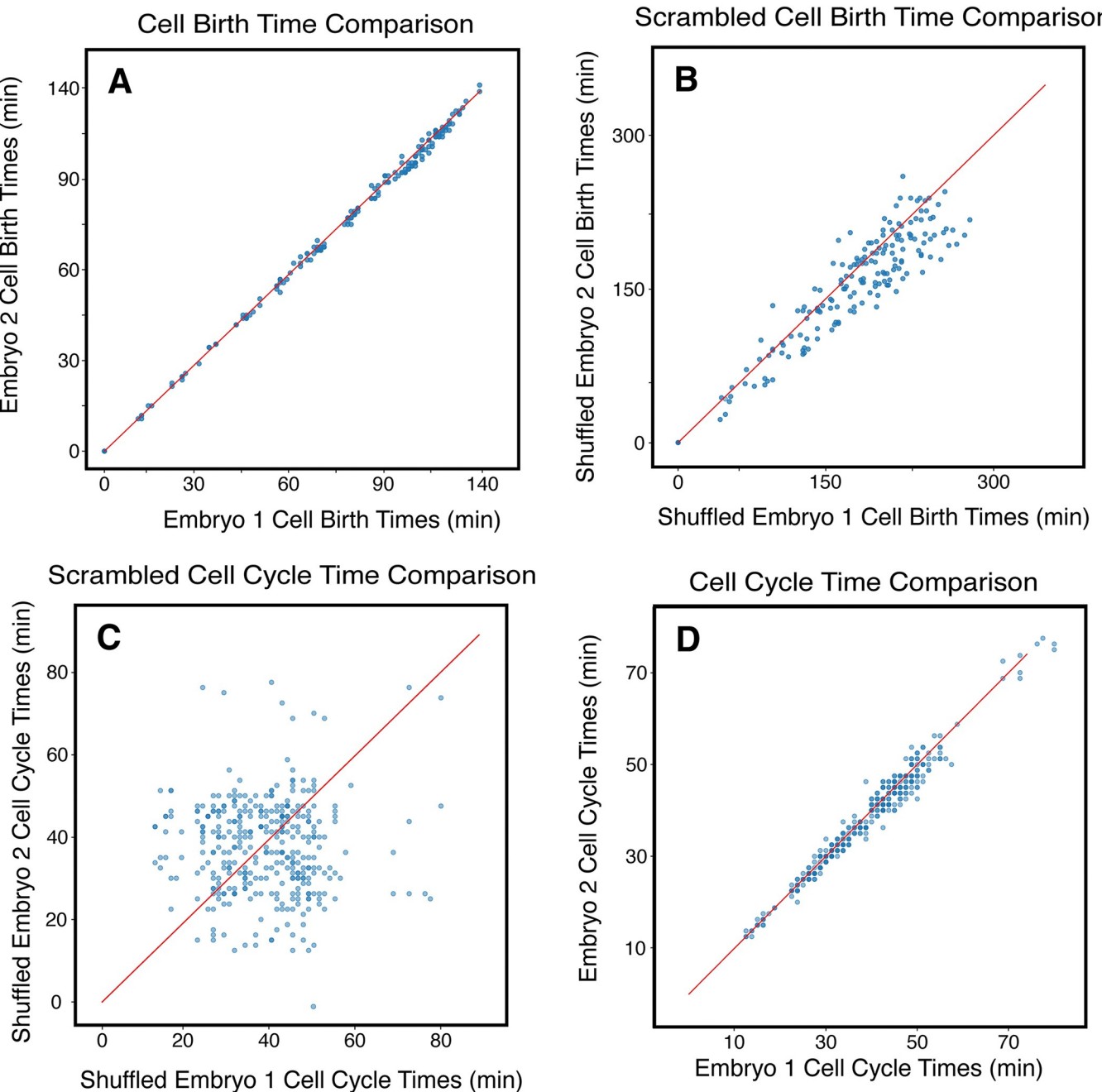

**Fig 2. Summation of Cycle Times into Birth Times Suppresses Variation.** (**A**) A comparison between the birth time of each cell (calculated as the sum of cell cycle times of each cell's ancestors) in two wild type *C. elegans* embryos. (**B**) Comparison between birth times calculated from two randomly shuffled wild type embryos, where each cell is assigned another random cell's birth time from within the lineage of the same embryo. Note that a significant correlation in birth times exists even in this shuffled data. (**C**) Comparison between shuffled cell cycle times rather than birth times. In this case, there is no correlation, as would be expected. (**D**) Comparison between the cell cycle times of each cell in two wild type embryos. Note that the same two embryos were used for all comparisons in panels A-D.

less than we observe with birth times. Since using the cycle time avoids spurious correlations and reveals more variation and structure in the data (Fig 2D), we focused on using the cycle time to calculate branch distances in this work.

We used the cycle times to calculate the branch distance between each pair of 30 WT embryos with cycle times taken from lineage tracing data from Du *et al.* [10] We then hierarchically clustered the embryos based on these distances. Surprisingly, clustering on the branch distance revealed two distinct previously unknown populations of WT embryos in the published dataset, representing differences in cell cycle timing between these two groups (Fig 3A). If we calculate the $R^2$ value between each pair of embryos, the two different clusters vanish (Fig 3B). To understand how these embryos could be highly correlated and still cluster into two groups based on the branch distance, we considered not just the correlation of the birth time relationship, but also its *slope m* (see Methods). Intuitively, this slope quantifies the systematic variation in developmental timing between two embryos and can be thought of as the slope of the best-fit line to the data in Fig 2D. When this slope "*m = 1*", it means that on average, each cell has similar cycle times between the two embryos; if the slope "*m < 1*", that means that on average, cells in the embryo on the x-axis have cycle times that are systematically *longer* than cells in the embryo on the y-axis (Fig 3C). We can interpret changes in this slope as representing changes in the relative "global clock" that times cell divisions in the two embryos. We calculated this slope for each pair of WT embryos in the data (Fig 3C). Note that here the embryo on the x-axis of the heat map is also used for the x-axis of the slope calculation. In this case, we used Principal Component Analysis (PCA) rather than linear regression to estimate the slope, since in any given comparison between embryos the choice of dependent vs. independent variable would be arbitrary (see Methods). These slope calculations, along with the high correlations in Fig 3B, indicate that the primary difference between these two groups of embryos is indeed the global rate of development. In particular, the larger group of "Cluster 1" embryos develop systematically slower than the "Cluster 2" embryos. This effect is unlikely to be a result of temperature differences as some embryos in Cluster 1 were imaged at the same time as some embryos that were found to be a part of Cluster 2. Some other epigenetic factor, such as a maternal effect, may have been responsible for this difference between the two populations [24]. This analysis exemplifies how the branch distance can reveal systematic differences in the data, such as this batch effect, that a focus on correlations alone cannot identify (Fig 3). As such, the Branch Distance provides a new graph-based metric that can identify differences that regression analysis alone cannot.

## The branch distance reveals heterogeneity between RNAi replicates

We then computed the tree edit and branch distances between the 1352 embryos treated with RNAi against 204 genes described by Du *et al.* [10] We hierarchically clustered these embryos based on the union branch distance into 4 major groups (Fig 4A), where the number of partitions was decided by analysis of the union branch distance dendrogram (S1 Fig). Of these, 2 clusters shown in the upper right corner and lower left corner of Fig 4A likely represent many outliers, as these embryos are approximately as different from one another as they are from the other 2 groups. Even among the remaining 2 clusters, we observe a significant degree of heterogeneity (S2 Fig). This heterogeneity exists not just between embryos treated with RNAi against different gene targets, but also between embryos treated with RNAi against the same gene (Fig 4B). The examples in Fig 4B highlight just two patterns that we observed. In the case of embryos treated with RNAi against *suf-1*, three pairs of embryos exhibit distinct levels of divergence from wild type lineage topologies (as indicated by the tree edit distance, Fig 4Bi) as well as from wild type patterns of cell cycle timing (as indicated by the branch distance, Fig 4Bii). RNAi against *skr-2*, on the other hand, induces minor defects in lineage topology (Fig 4Biii) but a broad spectrum of defects in the distribution of cell cycle times (Fig 4Biv). Surprisingly, this variability isn't a simple manifestation of variable phenotypic severity, as these

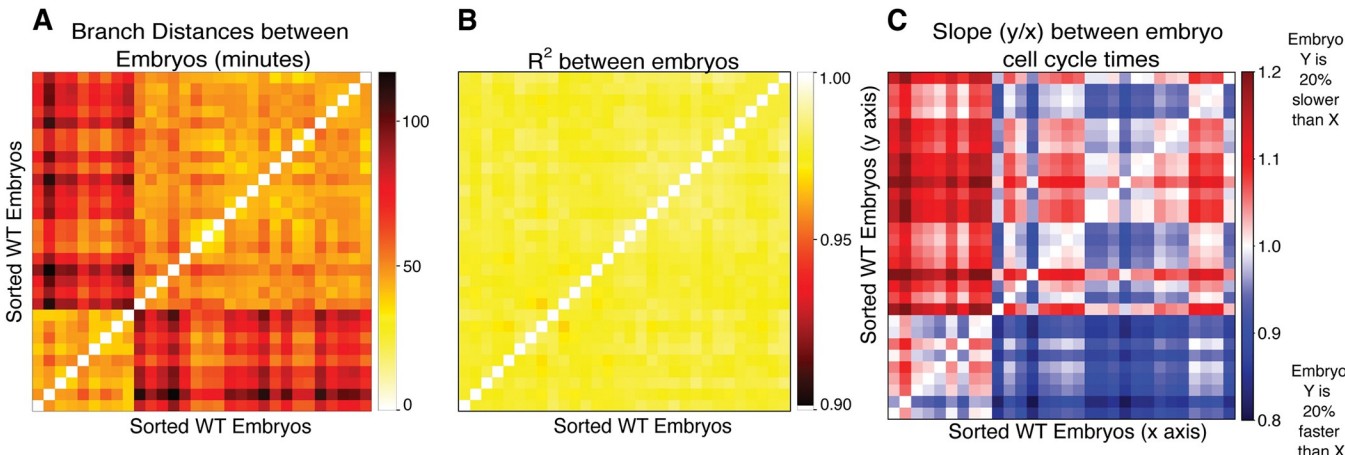

**Fig 3. The Branch Distance Reveals previously undetected Batch Effects in WT Embryo Cell Cycle Timing. (A)** Heatmap showing the union branch distance calculated between each pair of wild type embryos in the dataset. The ordering of embryos was sorted based on their assignment to two clusters computed using hierarchical clustering. **(B)** Heatmap showing the $R^2$ in cell cycle times between all pairs of WT embryos, sorted as in (A). **(C)** The slope calculated between cell cycle times between all pairs of WT embryos, sorted as in (A).

embryos often differ from one another as much as they differ from wild type, as embryos with the same gene knocked down are not necessarily in the same cluster (S1, S2, and S3 Datasets and S2 Fig). Prior work has demonstrated that some mutant phenotypes manifest variable penetrance due to underlying variation in endogenous gene expression [25], which may contribute to the variability we observe in combination with embryo-to-embryo variation in RNAi penetrance. We note that the wide degree of heterogeneity shown illustrates that the notion of "RNAi penetrance" as a continuous variation in the efficacy of gene knock-down does not translate to a similar interpretation of phenotypic severity, as downstream effects can be complex and heterogeneous.

Our findings in Fig 4A and 4B show that perturbations through RNAi can impact both lineage topology and the timing of cell cycle events. To separate these effects, we chose a single representative WT embryo from Cluster 1 (Fig 3A) and used this embryo to calculate both the tree edit distance between each RNAi embryo and WT and the intersection branch distance between each RNAi embryo and WT. We chose to focus here on the intersection branch distance because it focuses on just the duration of the cell cycle events among cells that are present in both the WT and RNAi-treated embryos; the union branch distance reflects both changes in timing and topology (Fig 1). In Fig 4C, we plot the tree edit distance to a WT reference embryo for each RNAi-treated embryo on the x-axis, and the intersection branch distance to WT on the y-axis. It is immediately clear that there is a bimodal distribution of tree edit distances, with a smaller subset of RNAi embryos having WT-like lineage topologies (with tree edit distances near 0) and most RNAi perturbations having a large impact on the structure of the lineage. Interestingly, we see that there is a general lack of correlation between tree edit distance and intersection branch distance, indicating that some RNAi perturbations have a large impact on topology, but the duration of the cell cycle is similar to WT amongst lineages with preserved topologies, while other perturbations leave the topology of the lineage almost intact but have a relatively large impact on cycle duration (Fig 4C).

We then examined whether RNAi against genes with related functions generated similar phenotypes based on our graph metrics. We grouped RNAi embryos together based on their functions as annotated by Du *et al.* [10] and observed a weak correlation between tree edit distance and intersection branch distance relative to WT (S3 Fig) although for most groups of

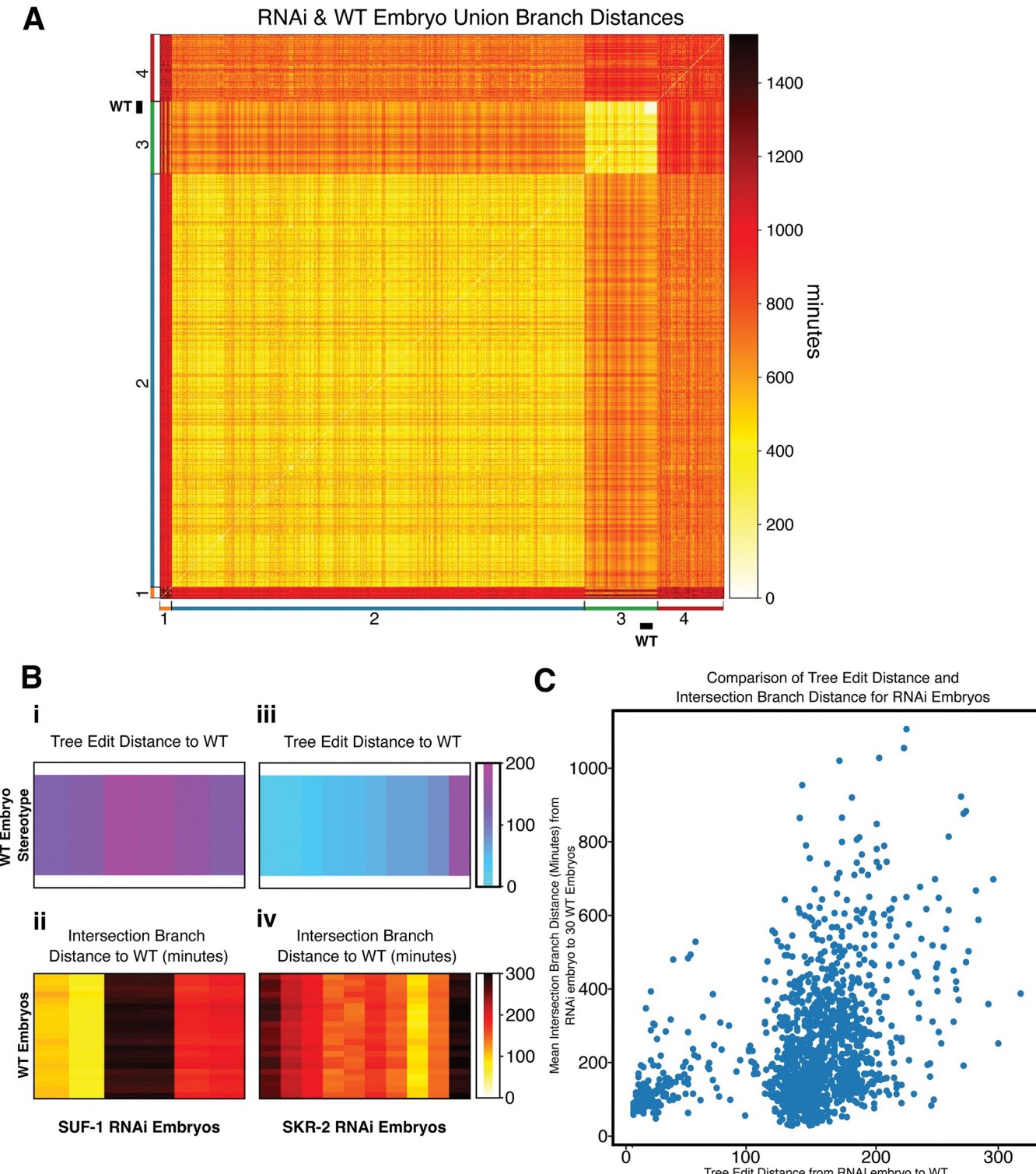

**Fig 4. The union branch distance reveals Heterogeneity in RNAi Cell Cycle Timing Coordination. (A)** Heatmap showing the union branch distance between all 30 WT embryos and 1322 RNAi Embryos in the dataset. Embryos were hierarchically clustered and sorted into 4 clusters shown along the axes of the heatmap, with WT embryos visible as a "white block" in cluster 3. **(B) i.** Distribution of the tree edit distance between 6 SUF-1 embryos and 30 WT embryos. **ii.** Distribution of the intersection branch distance between 6 SUF-1 embryos and 30 WT embryos. **iii.** Distribution of the tree edit distance between 10 SKR-2 embryos and 30 WT embryos. **iv.** Distribution of the intersection branch distance between 10 SKR-2 embryos and 30 WT embryos. **(C)** Comparison between the tree edit distance and intersection branch distance for each of the 1322 RNAi embryos relative to a single WT reference embryo.

genes, intra-class variability is greater than inter-class. We then asked whether any genes or functional classifications might be enriched in each cluster, finding that transcription factors, kinase/phosphatase, signaling and polarity related genes are overrepresented in the cluster nearest to the WT, while DNA replication/repair and mitochondria/stress genes are overrepresented in a cluster of highly heterogeneous embryos that are also the most distant from the WT embryos (S1 Fig). This is consistent with the findings of S3 Fig, which illustrates that kinase/phosphatase, signaling and polarity related genes are topologically and temporally closest to the WT, while DNA replication/repair and mitochondria/stress genes are furthest. While we cannot rule out the possibility that a portion of the observed heterogeneity among RNAi treated embryos may be a consequence of environmental or epigenetic factors as we found among WT embryos in Fig 3, the differences between WT clusters are dramatically smaller than the typical differences between RNAi treated embryos and at least a subset of treatments appear to cluster together reliably.

## Application of the branch distance to sublineages in WT and RNAi embryos

In all the work above, we applied our metrics to the cell lineage of entire embryos. While informative, this approach ignores the fact that certain developmental processes are specific to certain sublineages and might be lost in a global analysis. For instance, previous work on developmental timing in *C. elegans* focused on cell-by-cell comparisons and found that while cell birth times were globally well correlated [16], the specific ordering of cell divisions within the AB lineage was variable [26]. Given these observations, we wondered whether any structure existed in the distribution of cell cycle durations within the sublineages of each individual embryo.

We computed the intersection branch distance, which does not reflect differences in lineage topology, between each pair of canonical founder lineages in the early *C. elegans* embryo (Fig 5A). Cells in the *C. elegans* embryo are named based on their lineage history. A few founding cells in the early embryo possess unique names, but all cells derived from these are incrementally named according to the body axis it was born along. Cells with names containing the same number of characters following the unique name of the originating cell in the early embryo are thus born in the same generation of cell divisions and cells whose name only differs in the last character are siblings. The distribution of the intersection branch distance between each of the major differentiated lineages of the embryo show consistent patterns across the wild type samples that match the intuitive prediction that the posterior mesodermal and endodermal lineages derived from P1 are quite different from each other and from the AB lineage. We also found the same patterns reflected in the union branch distance which is sensitive to differences in topology and statistically significant differences (S4 Fig) between every pair of lineages derived from AB except for ABplp and ABprp. ABa and ABp derived lineages show distinct patterns of similarity, with the two lineages rooted at ABpl and ABpr being closer to one another than ABal and ABar, reflected in the left/right symmetric pattern of similarity in the lineages rooted at ABpxx and the lack of any such symmetry in ABaxx lineages. What is the origin of these patterns in cell cycle timing among the AB-derived sub-lineages?

## Notch signaling modulates cell cycle duration in a lineage-specific manner

Since Notch is responsible for breaking fate symmetry between the ABa and ABp lineages in the 4-cell embryo [27,28] and the pattern of branch distance distributions between the wild type AB lineages align with known Notch signaling events in the early embryo [29] (Fig 5B and 5C), we were interested in whether this pattern might be generated by Notch signaling.

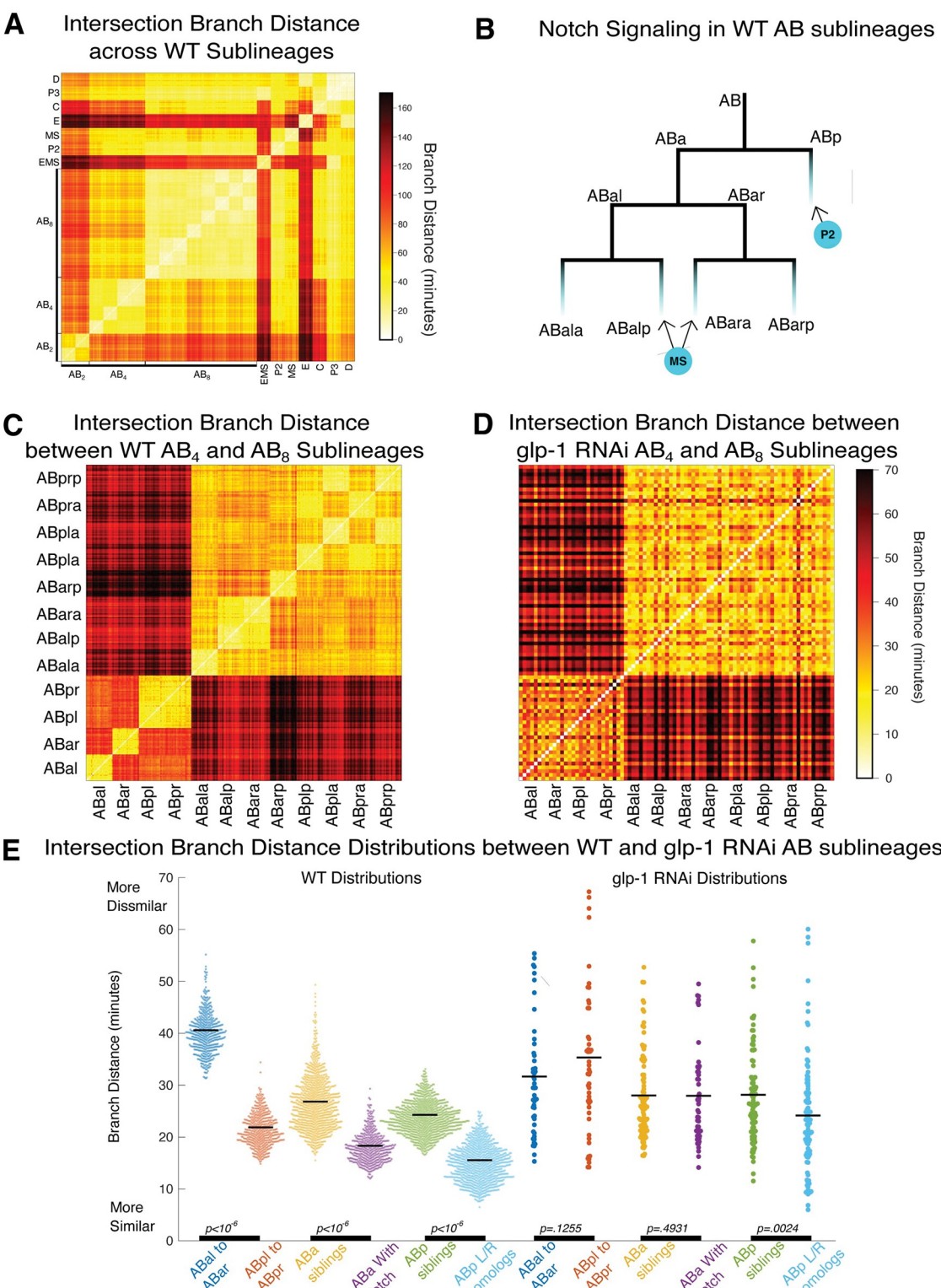

**Fig 5. Branch Distance reveals structure in the AB lineage. A)** Heatmap showing the intersection branch distance between every pair of sublineages in every pair of 21 wild type embryos. **B)** Illustration of the first two Notch signaling events in the early AB lineage. **C)** Heatmap showing a zoomed in view of the intersection branch distance between the 21 wild type embryos for each pair of AB-derived sublineages. Colormap is scaled from 0 to the max intersection branch distances between same-generation AB sublineages. **D)** Heatmap showing a zoomed in view of the intersection branch distance between AB-derived sublineages of 6

embryos treated with RNAi against *glp-1* for each pair of AB-derived sublineages. Colormap is scaled from 0 to the max intersection branch distances between same-generation AB sublineages. **E)** Distributions of intersection branch distances between subsets of AB-derived sublineages in WT embryos and embryos treated with RNAi against glp-1. P-values calculated using $10^6$ iterations of a permutation test.

The role of Notch in the patterning of C. elegans developmental timing has been studied in respect to asynchrony in the timing of divisions of sister cells [11]. The patterns we observe in the WT lineage based on the branch distance suggest that this may be a broader phenomenon affecting timing patterns throughout entire lineages. In *glp-1* RNAi embryos, the structure visible in the intersection branch distance between AB sublineages in wild embryos is clearly lost (Fig 5D). At the $AB_4$ stage, the two left/right symmetric lineages produced by ABp, which received a Notch signal from P2, are closer to one another by the branch distance than the two lineages produced by ABa are to each other (Fig 5E). Lineages derived from cells that independently receive Notch induction (ABalp and ABara by MS) also have a smaller branch distance between each other than between their direct siblings (Fig 5E). Embryos treated with RNAi against *glp-1* lose these differences and *glp-1* RNAi produces a nearly uniform pattern of branch distances among the AB lineages. Both ABp derived lineages in the wild type and all AB derived lineages in *glp-1* RNAi embryos exhibit a distinct pattern where left/right homologous lineages are closer to each other based on the branch distance than sibling lineages (Fig 5E). Perhaps Wnt signaling, which has been shown to incrementally accumulate in the posterior child of each cell division [30], continues to act to break fate symmetries between sibling cells in *glp-1* depleted embryos. Does the decreased intersection branch distance between Notch-stimulated sublineages represent a consistent effect on the duration of the cell cycle or increased variability between cognate cells across sub-lineages? To answer this, we compared the overall clock speeds of the $AB_8$ lineages, indicating that Notch affected lineages were faster compared to unstimulated lineages (S5 Fig) suggesting a consistent impact on cell cycle durations within sub-lineages, even when the stimulated sublineages are derived from separate founder cells that independently receive Notch stimulation.

## Preservation of cell cycle timing structure through lineage transformations

A key process in the early embryo is the differentiation of cell lineages needed for the formation of different organs and body parts. The large-scale RNAi screen performed by Du *et al.* systematically explored these phenomena using genetic markers of tissue fate [10]. They characterized diverse genes whose depletion results in homeotic transformations, where some cell lineages adopt the pattern of tissue fates normally produced by another lineage, and genes whose loss results in patterns of tissue fates not normally seen in any wild type lineage. Most of the lineages in the wild type embryo have both qualitatively and, as we showed above in Fig 5, quantitatively distinct patterns of cell cycle times.

We wondered whether patterns in cell cycle timing are a product of the same differentiation processes that define the tissue fate of these lineages. In other words, in an embryo where one lineage adopts the fate of another, does the pattern of cell cycle lengths in the transformed lineage change to match the pattern of the newly acquired fate? This question builds on a prior study which found that differences in cell cycle time between pairs of sibling cells indicated perturbations to cell fate in specific lineages [11]. We wondered whether lineage-wide transformations in fate might be detectable as a shift in cell cycle timing to match the pattern normally expressed by the acquired fate using the union branch distance. We designed a heuristic based on the branch distance to search for cases where this is true. For each homeotically transformed lineage identified by Du *et al.* [10] we refer to the transformed lineage as the

*origin* and the acquired lineage fate as the *destination*. To account for natural variation in each of the wild type lineages, we first define a diameter $D$ equal to the maximum pairwise branch distance between wild type examples of the *destination* fate (Fig 6A). We then assigned a transformation score to each *origin* lineage based on how many of the wild type *destination* lineages lie within $D$ $D$ minutes of the *origin* sub-lineage in any particular RNAi embryo (Fig 6B) and normalizing by dividing by the number of WT embryos (n = 21) used in the analysis. While most lineage transformations do not adopt the pattern of cell cycle times normally expressed by the acquired fate, we identified 95 cases where they do. Interestingly, cases where the transformed lineage falls within the neighborhood of all 21 wild type examples of the destination fate are more common than cases where the transformed lineage is proximal but not fully overlapping the neighborhood of the destination fate.

One advantage of using cell cycle timing and the branch distance as a phenotypic marker of lineage identity is that it can be assessed even in the absence of visible markers of cell fates, so we generalized our approach to measure the frequency of transformations between all possible pairs of lineages (Fig 6C). In this case, we find the nearest neighbor among possible *destination* fates for each *origin* lineage and count the number of wild type examples of the *destination* that fall within $D$ minutes of the *origin* lineage in the RNAi treated embryo. For genes with or without homeotic transformations identified on the basis of marker gene expression by Du *et al.* [10], the majority of *origin* lineages fall outside the range of variation of all wild type *destination* lineages, suggesting that the patterns of cell cycle times in the wild type lineage are very sensitive to genetic perturbation. In both figures, the two most populated bins are the bin with 0 WT neighbors followed by the bin with 21 WT neighbors (Fig 6B and 6C) noting that RNAi lineages aggregate more closely region around WT neighborhoods than regions further away (S6 Fig). Still, we identified 12 genes for which RNAi generates homeotic transformations based on both marker expression by Du *et al.* [10] and our approach of using the branch distance. These genes belong to a small set of key pathways that have well-known roles in specifying cell fate in the early embryo including Notch, Wnt, PAR polarity genes, and the maternally derived transcription factors *pie-1* and *skn-1* (Fig 6D). These pathways operate to break symmetry in the early embryo. This set is likely an underestimate since we have not accounted for two common types of perturbations to cell cycle timing and lineage structure: changes to the "global clock" (since the branch distance is not scale invariant), and the partial transformation of lineages since we examined only lineages rooted at the major founder cells in the early embryo. The nature of these transformations is consistent with prior work analyzing the change of cell cycle timing in the daughters of specific Notch receptors [11]. Specifically, each Notch induced homeotic transformation produces downstream timing perturbations that shift the entire transformed lineage to match the acquired fate.

Interestingly, examining each RNAi lineage on an embryo-by-embryo basis reveals striking diversity in penetrance and phenotypic consistency in homeotic transformations detected by Du et al. [10] (Fig 6D) and previously uninvestigated lineages (S4 Dataset), which list WT neighbors for RNAi lineages without annotated marker-based transformations. Several genes, such as *wwp-1*, *pop-1*, and *skn-1*, have sublineages that are within 1 diameter of their original and acquired fates, suggesting a degree of mixture in the neighborhoods of the third-generation descendants of the AB lineage. Furthermore, the variance in the number of embryos transformed and relative strength of each transformation suggests that RNAi penetrance and phenotypic severity are separable phenomena. In our case, we use penetrance to refer to the number of RNAi lineages that are transformed to the neighborhood of at least 1 WT reference lineage, while severity correlates with our measure of the transformation efficiency for each transformed lineage, a reflection of how close each transformed lineage is to the set of all WT reference lineages. For example, Notch pathway components *apx-1*, *glp-1*, and *lag-1* all have

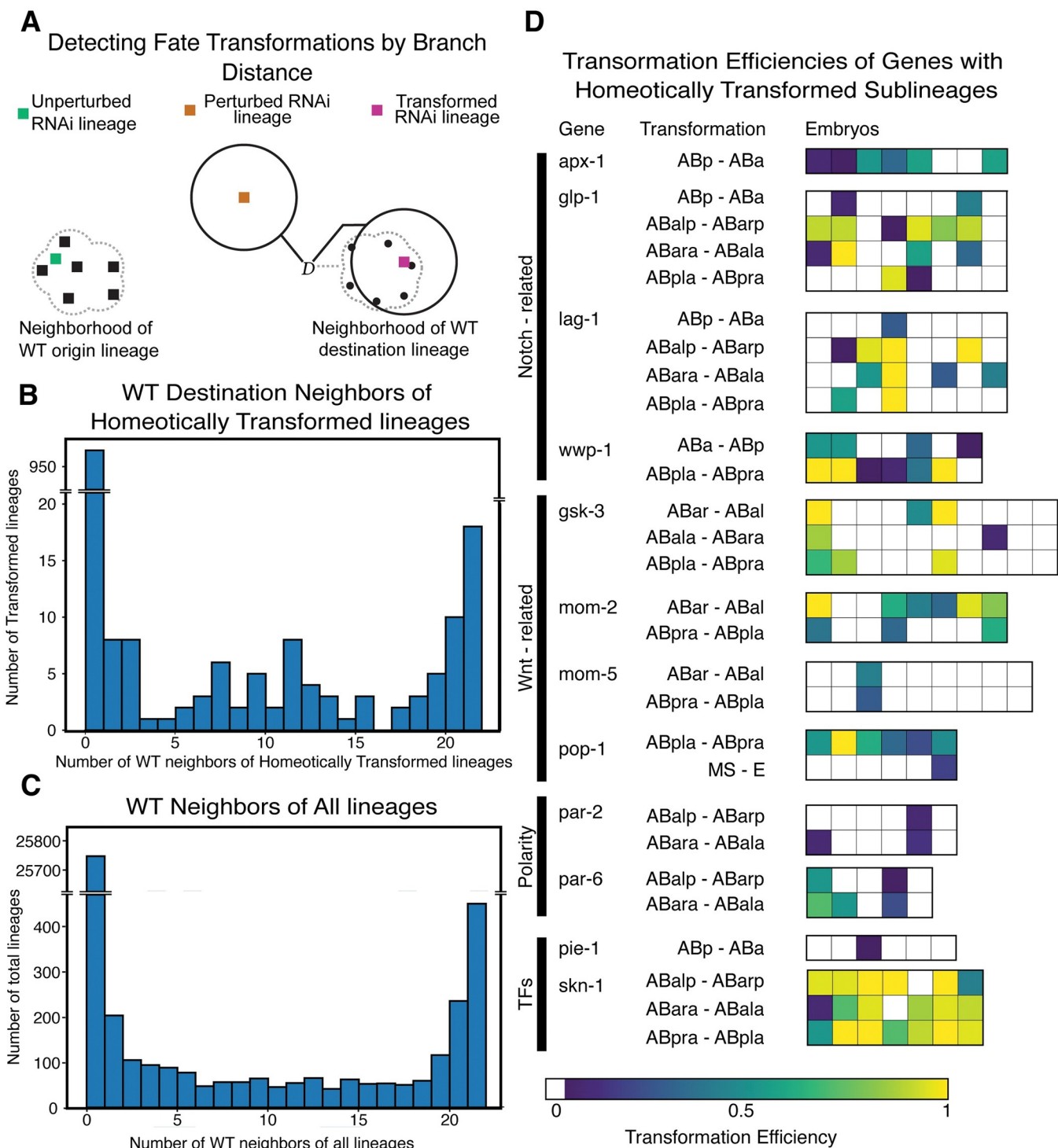

**Fig 6. Evidence for cell fate control over cell cycle timing. A)** Illustration of the transformation heuristic. For each WT destination lineage (black dots) a diameter $D$ is calculated as the maximum intragroup intersection branch distance. The transformation efficiency is then defined as the fraction of WT destination lineages that fall within diameter $D$ of each RNAi origin lineage (colored squares). In some cases, the transformation efficiency is 0 but the RNAi lineage has WT origin neighbors (green square) suggesting that the RNAi perturbed lineage maintained its original fate in terms of cell cycle timing. In other cases, this value is 0 and the RNAi origin lineage has no WT neighbors (orange square) suggesting that the RNAi perturbed lineage has both lost its original fate and failed to acquire the pattern of cell cycle timing of the destination lineage. In a minority of cases the RNAi origin lineage is within $D$ of 1 or more WT destination sublineages and a transformation efficiency is reported (magenta square). **B)** Histogram of the number of WT destination neighbors that homeotically transformed RNAi lineages have, using the heuristic defined in **A**. **C)** Histogram of the number of new WT neighbors that perturbed RNAi

lineages have. **D)** Heatmaps representing the transformation heuristic in **A.** for homeotically transformed lineages with at least 1 WT destination neighbor. The genes that induce these transformations and functions are listed alongside the corresponding heatmap of transformation.

transformations from ABp to ABa, where the transformed lineage lies within 1 diameter of a similar number of wild type examples. The more consistent presence of ABp to ABa transformations in *apx-1* might imply more complete penetrance of *apx-1* RNAi than that of *glp-1*, and *lag-1*, or that the knockdown of *apx-1* more consistently induces temporal transformations, suggesting a more central role for *apx-1* in fate specification for first-generation AB lineages. In contrast, for the E3 ligase *wwp-1*, the penetrance of the ABa to ABp transformation is both fairly high and occurs with a higher degree of transformation (as indicated by the fraction of wild type ABp sublineages within 1 diameter of the transformed lineage), or the transformation from MS to E in *pop-1* RNAi which is only observed in 1 embryo but is a perfect match for the neighborhood of all wild type E lineages, suggesting an extremely precise transformation of identity.

## Generalizing the branch distance to unlabeled binary trees reveals robustness in WT lineages

The principal novelty of our work, the branch distance, operates on labeled binary trees which allows one to trivially determine whether a node in one exists in another tree, and what the corresponding weights are. Node labels allow for easy comparison of corresponding nodes between trees, as without it, a space of alignments must be considered in order to calculate a distance. In the case of the generalized tree edit distance, the minimum distance of all possible alignments is considered [19]. Computing the branch distance between all possible alignments of two binary trees, selecting the smallest distance computed yields a simple generalization of the branch metric that preserves its desirable properties.

The value of a generalized branch distance would be its applicability to non-invariant lineages, yet few large scale datasets exist for species beyond *C. elegans* that include lineage topologies and phenotypic measurements for a large number of individuals. Thus, we benchmark this generalized metric using the *C. elegans* AB lineage. We applied the generalized branch distance between WT *C. elegans* $AB_8$ lineages, disregarding cell identity labels while iterating through all possible alignments and reporting the minimum computed branch distance. In comparing the pattern of distances produced by computing branch distances based on cell identity alignments (Fig 7A) to the pattern produced by computing generalized branch distances (Fig 7B), we find a high degree of similarity, demonstrating that the generalized branch distance is able to capture the same structure we found by computing the branch distance based on alignments that follow the *C. elegans* lineage.

If the minimum distance between all possible alignments is similar to the lineage-based alignment, we then asked whether the minimized alignment is itself the lineage-based alignment. To do this, we counted the percentage of cells in the embryo which have different matches between the alignment that produces the minimum branch distance, and the lineage-based alignment (Fig 7C). For pairwise alignments between lineages that have short branch distances between them, we find that the minimized alignment matches the lineage alignment very closely.

## Methods

Wild type and RNAi treated embryonic lineage data was retrieved as text files with each row corresponding to an individual cell in the lineage tree. Lineage relationships were

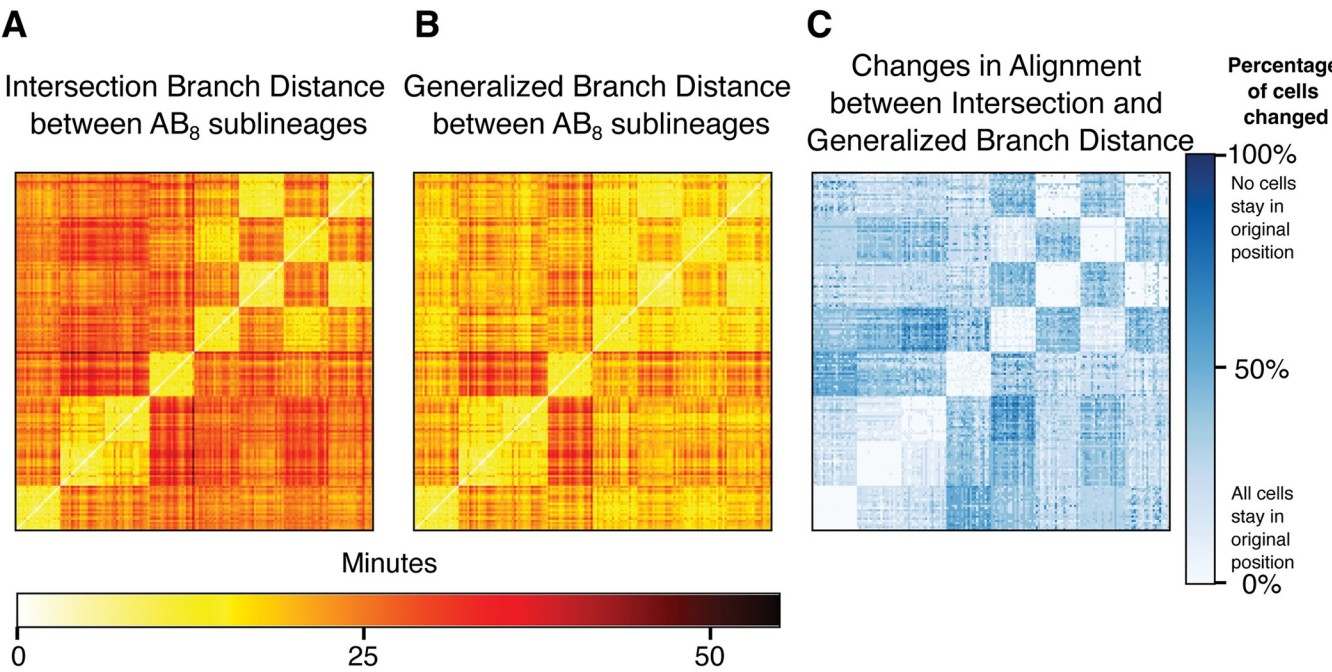

**Fig 7. The Generalized Branch Distance recapitulates structure in lineage coordination.** (**A**) Heatmap showing the intersection branch distance calculated between pairs of wild type $AB_8$ lineages (**B**) Heatmap showing the generalized branch distance calculated between pairs of wild type $AB_8$ lineages (**C**) Heatmap showing the percentage of cells in the lineage of one embryo that map to cells in a different position in the lineage of another embryo under the alignment produced by the generalized branch distance in (B).

reconstructed from the cell names, which are structured according to a common convention where a unique root cell ID indicates the identity of the founder cell of the lineage and each subsequent pair of cells is named according to the body axis along which its division was polarized. Cell cycle duration was extracted based on the number of columns associated with each cell from the data provided by Du *et al.* [10] used to report tissue specific transgenic reporter signal intensity. Each column represents the intensity measure made for each timepoint of imaging where the corresponding cell existed, thus we calculate the duration of each cell's cell cycle as 1.25 minutes per time point, based on the imaging frequency reported. A total of 30 wild type and 1322 RNAi treated embryos were retrieved and time-resolved lineage trees were generated from these raw data.

While all wild type embryos covered a uniform set of cells, RNAi treated embryos were only partially curated by Du *et al.* [10] to validate reporter expression. In order to address these discrepancies, we truncated each of these lineage trees based on the time cutoffs provided alongside the raw data and wild type embryos were pruned similarly for distance calculations. Each WT lineage is truncated based on the annotations provided by Du et al. [10] with RNAi lineages following gene and lineage specific cutoff instructions. The implementation of our data import and pre-processing is available alongside a complete codebase implementing our distance metrics and analysis routines at (https://github.com/shahlab-ucla/graph_distances))

## Graph based distance metrics applied to C. elegans lineages

In order to store the lineage data for each embryo as a binary tree, we take advantage of the naming convention for each cell using a standard hash table (or "dictionary" in python) data structure. A cell would be stored in the dictionary with name/reference X (i.e. the cell's name) and an element value representing its cell cycle time. The children of X would have the name/

reference of X followed by suffix 'a', 'l', or 'd', representing anterior, left, and dorsal orientations of division respectively. This cell would have a corresponding sibling name/reference of X with a suffix 'p', 'r', or 'v' representing a posterior, right, or ventral division relative to its sibling. For instance, a cell in the data set might have the name "ABal." This represents a cell descended from the AB cell in the embryo, where it is the left daughter of the anterior cell of the first AB division. With all cells in the embryo following this convention in the dictionary, any cell and all of its ancestors can be referenced by looking at the cell name and truncating its suffix one letter at a time.

The tree edit distance is a metric defined by counting "the minimum number of node deletions, insertions, and replacements that are necessary to transform one tree into another" as a measure of topological distance between trees (Fig 1A). This can be applied to the dictionaries that we use as proxies for graph structures. If a tree has a specific node not in another embryo, then a corresponding node must be inserted into the lacking embryo as a descendant of the appropriate shared node to produce topologically identical trees. Thus, a single operation has taken place to transform the structure of one tree into another. This approach can be generalized to describe any tree-based topological differences, as discussed in S1 Appendix. Using the dictionary format allows us to take advantage of the naming convention. Any cell that is added contains information pertaining to the connection to its parent nodes, allowing for trivial checks of hierarchical and topological relationships. Indeed, this can be expressed further by noting that addition/subtraction operations of nodes can be represented by the absence/presence of nodes in one embryo that is not in the other. Extending this concept allows us to calculate the number of transformation operations as the number of nodes that are in one, but not both, dictionary sets. This means that tree edit distance between two dictionaries with nodes under the naming convention of Sulston *et al* [3]. is defined as the magnitude of the intersection set subtracted from the magnitude of the union set of the dictionaries (in other words, it is size of the symmetric difference between the sets of nodes). In terms of the python implementation, this is calculated as the length of the XOR set of cells between the two embryos. It is utilized in Fig 4B (Looking at SUF-1 and SKR-2 RNAi embryos tree edit distance to WT stereotype) and in Fig 4C (Tree edit distance from each RNA embryo to WT stereotype Plotted on x axis).

In order to compare the trees in terms of the division timings, we introduce the concept of the branch distance. We define the branch distance as the Euclidean or $L^2$ norm of a vectorized representation of each lineage under comparison. To generate the vector, cells within each lineage are aligned first on the basis of depth from the root cell of each lineage and then on the lineage name derived from division orientation. In other words, we determine the *components* of each vector in such a way that division times for one cell are always being compared to division times for that same cell in a different embryo. When we calculate the $L^2$ norm, the difference between the values ascribed to a cell in one embryo and the corresponding cell in another embryo is taken and squared. Summing up these values and then taking the square root allows for an extension of the Euclidean norm to these weighted graphs.

To compensate for alternate topologies, we computed one of two variants of the branch distance. The Intersection branch distance only computes the distance on the intersection set of cells contained in both lineages (Fig 1B), treating values that are not shared as absent from the comparison. It is used in Fig 3A to look at distances between all 30 WT embryos, and hierarchically cluster them into 2 groups, with the larger group of 21 embryos representing the Wild Type in all future calculations (unless otherwise noted). It is also used in Fig 4B (Looking at SUF-1 and SKR-2 RNAi embryos intersection branch distance to 21 WT embryos), Fig 4C (Average branch distance from each RNAi embryos to 21 WT embryos Plotted on x axis), and Fig 7A (computing intersection branch distance between $AB_8$ sublineages).

Meanwhile, the Union branch distance treats any missing cells as having a cell cycle duration of 0 (Fig 1C). Thus, the Union branch distance compensates for differences in topology by directly adding the squares of values of cells without counterparts to the distance value, increasing it depending on topological variance and the value of the missing node. It is used to calculate the distance matrix between all WT and RNAi embryos (Fig 4A).

In comparing any two trees with any of these metrics, we note that the metric should work on subtrees or trees with different root nodes. This necessitates a change to the naming convention in cases where we compared different sublineages. This is done by finding the root node of both subtrees and assigning them an arbitrary letter. In cases where descendants of a root node are to be compared but have different orientations of division, we treat 'a', 'l', and 'd' suffix letters as equivalent as well as 'p', 'r', and 'v'. For example, the values of subtree ['A', 'Aa', 'Ap'] and the values of subtree ['B','Bl','Br'], if roots were normalized, would both have the naming convention ['Q','Qa','Qp']. Utilizing this convention allows us to apply the metrics described above to compute the distances between distinct sublineages. In Fig 5, this is used to compute the intersection branch distance between different sublineages in 21 selected embryos WT and all 7 *glp-1* knocked down embryos. In Fig 6, this is used to compute union branch distances between sublineages of 21 WT embryos and sublineages of RNAi embryos.

Our initial implementation of the branch distance assumes a unique alignment between the trees under comparison, as is enabled by the invariant lineage of C. elegans. In order to generalize this metric to cases where the optimal alignment is unknown, we adopt the approach utilized by the generalized graph edit distance [19]. Simply put, we compute the branch distance for all possible tree alignments and select the minimum computed value as the generalized distance. Note that there are several key constraints on this alignment. For one, we require that nodes only be aligned to nodes of the same depth in the other tree; a leaf node of one tree can't be aligned with the root node of the other tree. Secondly, if a node in one tree is aligned to a node in another, the child nodes must be aligned with one another as well. In other words, if we align node X from tree T1 to node Y in tree T2, then the child nodes of X can only be aligned to the child nodes of Y. This ensures that the alignments respect the topological structure of the tree. Proof of the generalizability of the branch distance is included in S1 Appendix. We thus apply the generalized branch distance to exhaustively search through all alignments of the WT $AB_8$ in Fig 7B.

## Calculating correlation of timing events between embryos

In previous work, some authors have used an alternative to the cell cycle time for comparing the timing of division events between embryos. Specifically, Bao *et al*. [16] compared embryos using the "cell birth time," defined as the time from the fertilization of the embryo to the birth time of a cell. It can be calculated as the sum of the cell cycle times of the ancestors of a cell. Previous authors have found extremely high correlations between different embryos using this birth time definition.

Since a cell's birth time is the sum of all previous division timings, comparing embryos using this parameter could suppress variation and introduce spurious correlations between embryos. A sum of random variables will often show less variation than the underlying variables themselves–this is the reason the "standard error of the mean" is generally less than the underlying standard deviation in the population. To test this, we shuffled all of the division times in the embryos in question. Specifically, we randomly assigned each cell in an embryo to the cell timing parameter of a different cell in the same embryo, effectively removing any correlation between the division timing of cells in any embryo while still preserving the underlying distribution of cell cycle times that can be produced.

A simple method of comparing the differences in cell timing events (Fig 2) is by plotting the times for each cell of one embryo against the times for each corresponding cell of another embryo. We then calculated the linear correlation coefficient between the cell cycle times between the cells of two embryos (Figs 2D and 3B) as well as the correlation coefficient between shuffled cell cycle times (Fig 2C). Shuffled birth times are computed by calculating the sum of the Shuffled cycle times of all ancestors of a particular cell and were also compared using the correlation coefficient (Fig 2B). Our analysis clearly demonstrates a significant correlation in cell birth times even in the shuffled data. As such, our subsequent analyses focused on comparing embryos using the cell cycle times.

## Computing the time between WT embryos

Our analysis in Figs 2 and 3A suggested that there are two distinct groups of WT embryos in the Du et al. data [10]. While the correlation between cell cycle times is lower than cell birth times (Fig 2), we nonetheless saw fairly high correlations between embryos of the two groups, despite their distinct branch distances (Fig 3A and 3B). We thus hypothesized that the difference between the two groups was due to a uniform rescaling of time–in other words, all of the division events in one group of embryos were likely slower than the events in the other group of embryos by a constant factor.

The plots in Fig 2 suggest a straightforward way to quantify this difference in timing: the slope of the timings in one embryo vs. another. If this slope is less than 1, this suggests that the embryo whose times are plotted on the x-axis develops slower than the one plotted on the y-axis; if the slope is greater than 1, that suggests the reverse. A natural way to estimate this slope would be to simply perform a linear regression between the two data sets. Doing so, however, involves selecting one set of timings as the "independent variable." Since both sets of timings in any comparison is subject to random variation, however, we chose a slightly different approach to calculating the slope.

To do this, we employed simple Principal Component Analysis (PCA) on each pair of embryos. The eigenvector corresponding to the largest eigenvalue corresponds to a line that best fits the principal axis of variation in the data. In all the embryo comparisons, this axis of variation corresponds naturally to the line that compares the cell cycle times between the two embryos (e.g., Fig 2D). We thus performed PCA on each pair of embryos with cell cycle times plotted against one another as in Fig 2D. The slope of this best fit line was then calculated by comparing the resulting principal eigenvector to the standard basis (i.e., calculating the "rise over run" for the eigenvector in the plane of Fig 2D). This method is used in Fig 3C to find the cell cycle scaling by comparing the cell cycle times of all 30 WT embryos against each other and partitioning the embryos into the clusters indicated in Fig 3A. These findings confirmed our hypothesis, indicating that the "group 1" embryos develop about 20% slower than the group 2 embryos.

## Clustering wild type and mutant embryos

We generated a distance matrix consisting of all pairwise union branch distances between WT and mutant embryos. We then performed single linkage hierarchical clustering on this distance matrix to generate a dendrogram between the embryos Since the number of clusters must be selected before the clustering is performed in hierarchical clustering, we analyzed the dendrogram to find a point with a large distance between generations [see S1 Fig for further details]. In the case of Fig 3, this approach partitioned the WT embryos into two groups. In the case of Fig 4, this approach resulted in 4 distinct clusters. Note that the distance matrices in Fig 5 were not clustered in order to show the pattern of variation between sublineages.

### Nonparametric permutation significance testing for distributions of distances

We found that the intersection branch distances between certain sublineages of WT embryos were generally smaller than the intersection branch distances between other sublineages. This difference seemed to be related to Notch signaling events during development (Fig 5A, 5B, and 5C). We used a simple permutation test to evaluate the statistical significance of this observation. In this test, we had two sets of distances: for instance, we compared the distances between ABal and ABar to the distances between ABpl and ABpr. This data corresponded to an observed difference between the means of the distribution. We then pooled the datasets together and generated randomized datasets of the same sizes by sampling without replacement. We calculated the difference of means between these randomized datasets. The p-values reported in Fig 5 represent the number of random cases where the absolute value of the difference in the means in these randomized datasets was greater than or equal to the observed difference.

### Detecting fate transformations for mutant embryos

Homeotic transformations that were identified by Du *et al.* [10] indicate cases where a sublineage of a RNAi embryo combinations of marker genes present more consistent with a *different* sublineage of the WT embryo. This transformation indicates that the RNAi treatment has transformed the cell-fate pattern from that typical of one sublineage (the "origin") into that of a different sublineage (the "destination"). A list of the transformations that take place is available at digital-development.org/download.html under the name "Excel spreadsheet of all homeotic transformation phenotypes". In our work, we considered whether these cell fate transformations also had an impact on cell cycle timing. To do this, we developed an approach to see whether one sublineage in an RNAi embryo was "close" to a different lineage in the WT embryos. Our approach represents the WT sublineage populations as point clouds in high dimensional space. Each such point cloud has associated with it the maximum distance between WT embryos in the population; we call this maximum distance the "diameter" of that sublineage (Fig 6A). In this case, we chose the larger group of 21 embryos with similar developmental timings that naturally cluster together in Fig 3A to avoid higher diameters that might arise from systematic differences in experimental conditions.

For any given RNAi embryo, we then compute the distance between each of its sublineages and each sublineage from each of the 21 WT embryos. Note that in this case we use the intersection branch distance. If the distance between an RNAi sublineage and a WT sublineage from *any* embryo is less than the diameter of the WT sublineage, we say that the RNAi sublineage is in the *neighborhood* of the WT sublineage from that embryo. If the RNAi sublineage is in the neighborhood of that same lineage in the WT embryos, then we say that the sublineage is "unperturbed." In other words, the origin and destination lineage for that sublineage is the same (in the sense of the neighborhood described above). If the RNAi lineage is in the neighborhood of a *different* lineage in the WT embryo, we say that that sublineage has been transformed (i.e., the origin and destination are different). If an RNAi sublineage is not in the neighborhood of any WT sublineage, then we say the sublineage is "perturbed" (Fig 6A). The degree of transformation/perturbation is quantified based on the number of origin/destination WT sublineages that are neighbors of each RNAi sublineage (Fig 6D).

Using the transformation framework outlined in Fig 6A, we then used a bootstrapping method to determine the significance of the distributions of Fig 6B and 6C. Specifically, we tried to determine whether lineages with randomly chosen cell cycle times lengths contain a number of WT neighbors comparable to homeotically transformed and other RNAi lineages.

In other words, we iteratively created "Null embryos" with all cell cycle times randomly assigned to a different cell (Fig 2) and computed the number of WT neighbors to the null embryos to find the degree to which completely randomized embryo times correspond with the Wild Type Embryo. In every homeotically transformed lineage, the length of every cell's cycle is shuffled among all cells of the same name across all embryos treated with RNAi against genes that produce homeotic transformations. For each of these shuffled lineages, the number of WT lineage neighbors (out of 21) is counted (S6B Fig). Along with homeotic transformations, we also shuffled all other RNAi lineages, where the length of every cell's cycle is shuffled among all cells of the same name across all embryos treated with RNAi, and counted the WT neighbors for each lineage (S6F Fig). We then repeated these shuffling 10000 times and counted the number of lineages with 21 WT neighbors (S6C and S6G Fig) along with the number of lineages with at least 1 WT neighbor (S6D and S6H Fig).

## Discussion

Analyses of the structure of lineages in biology have focused principally on the construction of phylogenies [14,15] and measurements of inter-node distances within individual trees [31]. This approach has been shaped by the requirements of taxonomic work, where no ground truth topology exists, and multiple measures of distance might be employed. Cell lineages, on the other hand, have clearly defined structure, and recent work has explored strategies for measuring differences in tree topology [18]. Recently, techniques from spectral analysis were applied to phenotypic measures aligned to cell lineages, including in *C. elegans*, but with an emphasis on characterizing these phenotypes in the context of lineages with variable structure [32]. Automated cell lineage tracing is an increasingly mature technology, having been applied to *C. elegans* [12,32,33] *Drosophila* [34], zebrafish [35], and mouse [36] development as well as to the study of lineage relationships in stem cell [37] and immune cell [38] culture systems. A limited set of metrics have been applied to the comparison of cell lineages [18,32], in part driven by the unordered nature of cell lineages reconstructed in most organisms. In the case of *C. elegans* lineages, for which there are now public repositories containing measured lineages from thousands of wild type and perturbed embryos [10,13,39], the ordered and stereotypical nature of its somatic lineage removes the need to align the lineages, thus dramatically simplifying the application of graph-based approaches to the problem of quantitatively comparing lineage trees. We thus applied the intuitive graph-theoretic notion of the tree edit distance [19], and its extension in the branch distance, to dissect the structure of *C. elegans* embryonic lineage. We note that the tree edit distance, when computed on labeled binary trees, is operationally identical to the Robinson-Foulds metric, as each of these quantifies the magnitude of the exclusive disjunction between elements in two trees [40]. These metrics allowed us to uncover previously unknown heterogeneity between populations of wild type embryos, to quantify the variability of RNAi-induced lineage phenotypes, and to shed light on key mechanisms of patterning in early embryogenesis, and expand on analyses of RNAi induced phenotypic changes [11,12]. Most prior analyses of developmental timing in the *C. elegans* embryo used the time of a cell's birth relative to an early reference point (ex. the first division of the zygote) and correlation between the birth time of the same cell across embryos as a measure of developmental similarity. We showed here that these two choices mask heterogeneity present in previously published records of wild type development. We wondered then whether these effects, combined with 1-to-1 comparisons of timing between the same cell across multiple embryos may have obscured patterns in developmental timing across lineages within the embryo.

Using the tremendous volume of existing lineage data available from the work of Du *et al* [10], we sought to benchmark our metrics on wild type and RNAi-perturbed embryos and

explore whether a lineage-centric view of developmental timing may reveal previously unappreciated patterns. It is well known that RNAi, especially by feeding in *C. elegans*, induces phenotypes with variable penetrance [41]. It has been shown that variable penetrance in mutants may occur due to underlying heterogeneity in gene expression [25]. Our graph metrics show that phenotypic variability under RNAi can exhibit a wide range of patterns of severity. This includes patterns that correspond to linear gradients of severity, multimodal distributions of phenotypes, and apparently random variation between individual embryos (Fig 4B). These measurements show that, for many genes, RNAi induces variability between individual embryos that is often on par with the phenotypic distance between wild type and individual RNAi embryos. Our approach expands on prior work which tracks changes through phenotypes in individual cells [11,12] noting that our approach allows for multimodal measurements, allows for rigorous distance-based bioinformatics analyses, and consolidates lineage descent data.

Taking advantage of the ordered nature of cell divisions in the *C. elegans* embryo to align arbitrary pairs lineages within the embryo, we sought to characterize the structure of cell cycle duration in the wild type lineage. We were especially surprised to find reproducible patterns within the lineages derived from AB, the larger of two cells born from the first asymmetric division of the zygote. In *C. elegans*, the major patterns of cell fate that are established by intercellular Notch signaling are well known, and the pattern of branch distances between the AB-derived lineages we observed aligns perfectly to the first two Notch signaling events in the early embryo. RNAi against Notch/*glp-1* abolishes this structure, demonstrating that this pattern of cell cycle timing in the AB lineage is a product of Notch signaling. Lineages that receive Notch signals also exhibit on average shorter cell cycle lengths than lineages that do not (S5 Fig).

Biophysical parameters such as cell volume affect cell cycle duration [42–46], but genetic regulation of subtle differences in cell cycle timing may occur via many potential mechanisms. Du *et al.* [9,10] demonstrated using transcriptional reporters of tissue fate that the loss of any one of many genes essential for development can induce homeotic transformations between the major founder lineages in the early embryo [9,10]. We set out to determine, using the branch distance, whether developmental timing in transformed lineages is independent of lineage fate, is transformed along with fate, or is lost upon fate perturbation. We devised a simple heuristic to assess the proximity of RNAi-treated origin lineages to the wild type destination lineage that expresses the closest pattern of cell fates as defined by Du *et al.* [10] by counting the number of wild type examples of the destination lineage that are less than the maximum inter-wild type branch distance away from the RNAi origin lineage. Using this conservative approach, which would fail to detect transformations in cases where the global embryo clock is altered or where subsets of individual lineages are transformed, we find that only a handful of genes (12 genes out of 204 characterized) induce homeotic fate transformations where developmental timing in the transformed lineages also transforms to match that of the newly acquired fate. This set is composed of genes in the Notch and Wnt pathways, two PAR polarity genes, and the maternally derived transcription factors *skn-1* and *pie-1*. The fact that most perturbations produce homeotic transformations generate patterns of cell cycle duration that match neither that of the original wild type lineage or the newly acquired fate suggests that lineage-specific developmental timing is likely quite sensitive to genetic perturbation. When homeotic transformations in cell cycle timing do occur, a perfect match to wild type lineages outnumbers incomplete matches suggesting that, despite its sensitivity to perturbation, the wild type patterns of cell cycle timing may represent stable states. This tool could be extended to any quantitative cell specific measurement in any tracked lineage, extending the definition and modalities used to measure cell fate. Our generalized formulation of the

branch distance expands this capability by allowing for its use in cases where no intuitive alignment exists between pairs of cell lineages, for example in the development of non-eutelic animals.

Our analysis demonstrates that the genetic identity of cell lineages can reproducibly and finely tune the distribution of cell cycle duration within cell lineages. It is interesting that the pathways that preserve lineage-specific developmental timing across homeotic transformations are known to play a critical role in cell fate specifications upstream of most tissue-specific transcriptional programs. It is thus likely that either a specific subset of factors downstream of fate regulators or finely tuned expression levels of tissue-specific genes are required for the proper patterning of cell cycle duration within lineages. Whether this tuning is itself a functional element of the developmental program remains unclear. Perturbations to key cell cycle regulators generate dramatic changes in cell cycle duration as well as homeotic fate transformations in *C. elegans* [47–50], and changes in the duration of the cell cycle of stem cells in other systems are correlated with specific cell fates such as in the generation of bipolar cells during retinal development [51]. Our results demonstrate a precise relationship between cell fate and developmental timing that motivates revisiting gaps in our understanding of links between cell cycle regulation and cell fate control. More broadly, our findings highlight the ways in which quantitative analysis of phenotypic similarity can reveal unexpected structure in animal development. In particular, the use of pairwise distance metrics applied to lineage-resolve metrics allows for an intuitive extension of notions of cell state and identity. Reducing these multidimensional data types using such intuitive measures of distance simplifies the application of common data exploration and visualization strategies.

## Supporting information

**S1 Fig. Structure in the distance between all WT and RNAi embryos.** Shown is a dendrogram constructed using the union branch distance measured between all WT and RNAi treated embryos along with the 4 classes we partition the dataset into. The WT embryos and WT-like RNAi-treated embryos are highlighted in cluster 3. Significantly Overrepresented Genes and Functional Classifications in each cluster are listed here. P values were calculated with Boschloo's test and significance was determined based on a Bonferroni corrected threshold of $1.5 * 10^{(-4)}$. Raw data are available in supplemental datasets S1 and S3.
(TIF)

**S2 Fig. RNAi embryos are far more dispersed than WT embryos. (A)** The distribution of union branch distances between wild type embryos (red) and between embryos treated with RNAi against the same gene (blue). Densities were generated using a kernel density estimator. **(B)** Strip plot of all RNAi embryos showing the branch distances between embryos treated with RNAi against the same gene. Embryos treated with RNAi against genes with shared functions are grouped together. The median branch distance within each set of embryos is plotted as a large circle.
(TIF)

**S3 Fig. Correlation between the tree edit distance and branch distance between RNAi treated embryos and a single WT reference embryo.** The mean (circle) and minimum/maximum values (blue lines) of the distance between embryos treated with RNAi against genes with common function and a single WT reference embryo are shown.
(TIF)

**S4 Fig. All but one WT sublineages possess unique patterns of cell cycle timing. (A)** A heatmap showing the union branch distance between each pair of sub-lineages across 21 WT

reference embryos. **(B)** Heatmap showing the p-value calculated using 441 permutation tests (see methods) against the null hypothesis that sublineages share a common distribution of intersection branch distances. All but a single comparison (ABprp vs ABpla) are significant based on a Bonferroni corrected threshold of $2 * 10^{-5}$.
(TIF)

**S5 Fig. Notch induction systematically shortens cell cycles in affected sublineages. (A)** A heatmap showing the fraction of cells in the lineage listed along the Y axis that have a shorter cell cycle than the corresponding cell in the lineage along the X axis. Each sublineage contains 32 cells so a lineage in which >16 cells possess a shorter cell cycle is considered to be "faster" **(B)** The distribution of comparisons from the heatmap in panel A grouped based on lineages that do not have a history of Notch activation (green, ABala and ABarp), lineages derived from ABa in which Notch is activated (red, ABalp and ABara), and all lineages derived from ABp.
(TIF)

**S6 Fig. Significantly more transformed lineages match the cell cycle timing of their ectopic fate than random. (A)** The number of WT destination lineages that fall within the transformed neighborhood of lineage annotated as homeotically transformed by Du *et al.* [10]. See Main Fig 6B. **(B)** The number of WT destination lineages that fall within the transformed neighborhood of homeotically transformed lineages where the length of every cell's cycle is shuffled among all cells of the same name across all embryos treated with RNAi against genes that produce homeotic transformations. Note there are no WT neighbors to any of these shuffled lineages. **(C)** The number of cases where lineages from RNAi treated embryos that are shuffled as in (B) fall within the neighborhood of all 21 WT samples of the destination lineage. The red line shows the corresponding value for the unshuffled data. **(D)** The number of cases where lineages from RNAi treated embryos that are shuffled as in (B) fall within the neighborhood of 1 or more WT samples of the destination lineage. The red line shows the corresponding value for the unshuffled data. **(E)** The number of WT destination lineages that fall within the transformed neighborhood of all sublineages from all RNAi treated embryos. See Main Fig 6C. **(F)** The number of WT destination lineages that fall within the transformed neighborhood of all lineages from RNAi treated embryos where the length of every cell's cycle is shuffled among all cells of the same name. **(G)** The number of cases where lineages from RNAi treated embryos that are shuffled as in (F) fall within the neighborhood of all 21 WT samples of any lineage. The red line shows the corresponding value for the unshuffled data. **(H)** The number of cases where lineages from RNAi treated embryos that are shuffled as in (F) fall within the neighborhood of 1 or more sample of any lineage. The red line shows the corresponding value for the unshuffled data.
(TIF)

**S1 Dataset. Number of RNAi embryos in each Cluster.** Text file listing the number, type, and p-value analyzing whether RNAi embryos with specific genes knocked down were significantly overrepresented in each cluster, from the hierarchically clustered union distance matrix (Fig 4A) between all embryos and its corresponding dendrogram (S1 Fig). P-values were calculated with Boschloo's Test.
(TXT)

**S2 Dataset. List of Genes and the corresponding clusters.** Text file listing each gene and the Clusters its corresponding RNAi embryo belongs to, from the hierarchically clustered union distance matrix (Fig 4A) between all embryos and its corresponding dendrogram (S1 Fig).
(TXT)

**S3 Dataset. List of clusters and significant functional classifications.** Text file listing each cluster, the functional classifications of all genes in the cluster, and an enrichment analysis analyzing whether each functional classification is significantly overrepresented in that cluster. P-values were calculated with Boschloo's Test (S1 Fig.)
(TXT)

**S4 Dataset. List of All Found Transformations.** Text file listing each Gene, its corresponding RNAi embryos and sublineages, along with listing each RNAi sublineages WT neighbor names and frequencies (Fig 6C).
(TXT)

**S1 Appendix. Proof that the generalized branch distance is a metric and comparisons between the tree edit distance and the Robinson-Foulds distance.**
(PDF)

## Acknowledgments

The authors would like to thank Dr. Zhou Du and Dr. Anthony Santella for guidance in parsing their lineage data, and Dr. Roy Wollman and Dr. Alex Hoffman for their feedback and advice, along with present and former members of the Deeds and Shah Labs.

## Author Contributions

**Conceptualization:** Eric J. Deeds, Pavak K. Shah.

**Data curation:** Pavak K. Shah.

**Formal analysis:** Gunalan Natesan, Eric J. Deeds, Pavak K. Shah.

**Funding acquisition:** Pavak K. Shah.

**Investigation:** Gunalan Natesan, Pavak K. Shah.

**Methodology:** Gunalan Natesan, Eric J. Deeds.

**Project administration:** Eric J. Deeds, Pavak K. Shah.

**Resources:** Pavak K. Shah.

**Software:** Gunalan Natesan.

**Supervision:** Eric J. Deeds, Pavak K. Shah.

**Validation:** Timothy Hamilton, Pavak K. Shah.

**Visualization:** Gunalan Natesan, Pavak K. Shah.

**Writing – original draft:** Gunalan Natesan, Eric J. Deeds, Pavak K. Shah.

**Writing – review & editing:** Gunalan Natesan, Timothy Hamilton, Eric J. Deeds, Pavak K. Shah.

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
