## [Decision Letter · Decision Letter 0]

18 Aug 2023

Dear %TITLE% Shah,

Thank you very much for submitting your manuscript "Novel metrics reveal new structure and unappreciated heterogeneity in C. elegans development" for consideration at PLOS Computational Biology.

As with all papers reviewed by the journal, your manuscript was reviewed by members of the editorial board and by several independent reviewers. In light of the reviews (below this email), we would like to invite the resubmission of a significantly-revised version that takes into account the reviewers' comments.

The reviewers identified numerous issues that should be addressed in a revised manuscript. It will be especially critical to address the reviewer comments regarding generalizability of the method and novelty over previous methods. 

We cannot make any decision about publication until we have seen the revised manuscript and your response to the reviewers' comments. Your revised manuscript is also likely to be sent to reviewers for further evaluation.

Sincerely,

Shaun Mahony

Academic Editor

PLOS Computational Biology

Pedro Mendes

Section Editor

PLOS Computational Biology

Reviewer's Responses to Questions

**Comments to the Authors:**

Reviewer #1: The authors in this interesting manuscript claim that they have introduced a generalized metric, branch distance, that allows the discovery of new variability based on phenotypic measurements in individual cells during the development of C. elegans embryos. My main concern with this study is 1) the lack of proven generalizability of the metric, as it seems to me that it can only for now be applied to organisms with a very well defined cell lineage such as C elegans. 2) Although the discovery of variability in the phenotype made by comparing close cells in the lineage tree (using their branch distance metric) using their phenotypic measurements (i.e gene expression and cell division time) are interesting, the authors need to prove that other used metrics such as RF or triplets would not give similar results. If such is the case, their approach would be generalizable to other situations were the lineage tree is not prototypic (the limited case of C elegans) such as for the development of other organisms where both lineage and gene expression can now be traced molecularly. If the authors are unable to show this, nor able to find the genetic origins of the correlations/diversity between cell division variability and cell lineage status, this study would be of restricted interest.

Reviewer #2: The authors introduce a generalized metric to compare lineage topology between wild type (WT) or RNAi embryos. They apply the metric to the data on cell-cycle timing from over 1300 WT and RNAi-treated Caenorhabditis elegans embryos. They claim that they found surprising heterogeneity within this data set, including subtle batch effects in WT embryos and dramatic variability in RNAi-induced developmental phenotypes. They also claim that heterogeneity have been missed in previous analyses. After further analysis of these results, they suggests a novel, quantitative link between pathways that govern cell fate decisions and pathways that pattern cell cycle timing in the early embryo. Overall, the branch distance they proposed has the potential to more accurately quantify the subtle variation/change in lineage topology than the existing ways of lineage quantification. At the same time, there are many issues that needed to be addressed before acceptance for publication.

Major comments

1. The authors claim that heterogeneity in lineage topology is missing previously. However, this claim seems not be able to hold. For example, impact of Notch signaling on cell cycle timing [https://doi.org/10.15252/msb.20145857], variability of cell lineage between wild type embryos [http://doi.org/10.1016/j.ydbio.2012.11.034] among others. They should articulate what the different finding they made compared with previous findings. These previous findings should be properly cited.

2. The method for calculating cell cycle times was reported previously [https://doi.org/10.15252/msb.20145857] and [https://doi.org/10.1101/776062], while it is well justified why the authors used cell cycle time instead of cell division time (although it might be of less importance since this has been used by other groups and for many years). This should be mentioned and critiqued in their introduction.

3. In Figure 3, the branch distance can tell the difference in the developmental pace between two clusters, but this has also been reported by the two pieces of literature above. So it’s a support for the authors’ method as well, not new finding.

4. In Figure 4, the authors reported the phenotypic heterogeneity within RNAi-treated embryos of the same gene and between RNAi-treated embryos of different genes, which was not unexpected, and the possibility of experimental artifacts (e.g., temperature control) cannot be excluded. It’s still difficult to interpret the clusters.

5. In Figure 5, the authors report how Notch signaling affects cell lineage. It would be better to putthe results in the context of what has been reported in [https://doi.org/10.15252/msb.20145857] and what novel finding is here.

6. In Figure 6, although a common genetic architecture controlling cell division asynchrony and cell fate asymmetry was reported, the idea of using cell lineage to predict cell fate is novel and interesting. It should be expanded more regarding its potential to uncover the underlying genetic functions.

Specific Comments:

1. Page 4, Line 71, “Aligning measurements such as…” —— Apart from the developmental properties mentioned by the authors, another well-known hallmark of C. elegans development is its stereotypical cell position [https://doi.org/10.1016/0012-1606(83)90201-4]. Aligning measurements and intuitive visualization has been done in [http://doi.org/10.1016/j.ydbio.2012.11.034] and [https://doi.org/10.1101/776062], which should be mentioned to give a more comprehensive description of stereotypical C. elegans development.

2. Page 4, Line 83, “This property dramatically simplifies the application of metrics computed on lineage trees…” —— [https://doi.org/10.15302/J-QB-022-0319] proposed several graph metrics for C. elegans embryonic cell lineage recently and their methods can also tell the difference between the differentiated ABp and EMS blastomeres, so it should be cited.

3. Page 5, Line 105, “We extended these measurements…” —— The role of Notch signaling in the control of developmental timing has been reported previously by [https://doi.org/10.15252/msb.20145857] using the wild-type and RNAi-treated embryos collected with similar experimental methods of the embryos used in this manuscript, but was ignored here. The paper reported the effect of lag-1, sel-8, ref-1, and lin-12/glp-1 (the gene specifically studied in this manuscript) on the developmental timing in Fig. 5. The authors should at least explicitly describe what is new they found compared to the previous knowledge in this highly relevant literature, and their Introduction, Results, and Discussion parts should be rephrased accordingly.

4. Page 6, Line 129, “The first is the tree edit distance…” —— The quantitative comparison between a pair of cell lineages proposed in this manuscript needs to determine the nodes at the beginning, and calculate the three values. The nodes are counted within a time window, but how to establish this time window for different cell lineages? For example, regarding the AB lineage, the wild-type embryos achieve 8 or even more AB cells at different timing, so the operation for picking up the time window must affect the calculation results, especially for the RNAi-treated embryos with diverse cell lineages. The authors should provide thes details in the main text or supplemental material to accommodate these problems.

5. Page 7, Line 160, “Fig 1…” —— The mathematical form of equations and symbols should be consistent across the whole manuscript, for example, now the L2 is italic on Page 7, Line 160 but all the mathematical symbols in Fig. 1 are not. Please also indicate that TED, IBD, and UBD are abbreviations in either the figure or its legend to avoid confusion.

6. Page 7, Line 160, “Fig 1…” —— The idea that uses graph metrics to compare any measurements on two cell lineages is interesting. The number noted near the branch is such an assumed measurement, like the cell cycle timing or gene expression level that has been emphasized in this manuscript. However, the figure presentation is a little bit confusing as the branch length is not based on its assumed cell cycle timing and the branch color is not consistent with its assumed gene expression level. The authors may want to present it in a more generalized way, but it is difficult to be understood by the audience. One suggestion is to expand the figure by showing how cell cycle timing or gene expression level is distributed on the cell lineages according to their given values or based on a real case (real cell cycle data and actual gene expression data).

7. Page 8, Line 180, “While the spatial organization…” —— The statement “the distribution of cell cycle times within lineages have not been as deeply explored” is inappropriate. There are two pieces of literature, [https://doi.org/10.15252/msb.20145857] and its subsequent update [https://doi.org/10.1101/776062], which calculated this distribution with similar methods (i.e., using cell cycle times instead of cell division times) and produced similar results (i.e., finding the high correlation between cell cycle times of different wild-type embryo samples) as reported in this manuscript. Those previous results should be mentioned for supporting/validating the data and method used in this manuscript and the subtitle “unexpected batch effects” should be revised/softened.

8. Page 12, Line 227, “Surprisingly, clustering on…” —— Why are there two distinct populations of WT embryos? Do they come from different experimental conditions [https://doi.org/10.1016/j.ydbio.2012.11.034]? If the embryo data is solid, there should be a complete record of this, and the authors should be able to explain why this happen and avoid any possible artificial/methodological mistakes. Otherwise, there might be something wrong in the data used in this manuscript. More importantly, this can help prove the power of the graph metrics.

9. Page 13, Line 249, “Fig. 4…” —— Which cluster is the wild-type embryos? Should they be separated as they are supposed to have the highest similarity?

10. Page 14, Line 261, “We hierarchically clustered these embryos based on the union branch distance into 4 major groups… ” —— As stated above, why can the RNAi-treated embryos be clustered? Do they have different known gene functions? As mentioned by the authors that the graph metrics can tell the difference between the developmental pace of wild-type embryos, is it possible this clustering results just come up because the original experimentalists have different experimental conditions such as a slight change in temperature? The authors should explain the reason or at least, make speculation instead of just drawing strong conclusions with the computational outputs.

11. Page 14, Line 274, “Surprisingly, this variability isn’t a simple manifestation of variable phenotypic severity, as these embryos often differ from one another as much as they differ from wild type…” —— This comparison of variability is insufficient to claim that it is not variable phenotypic severity. In theory, a multi-body system is sensitive to thermal noise, which is common in a biological system, then the system state would keep being diversified when the noise is not regulated/controlled ~ the so-called butterfly effect in nonlinear science. This has been demonstrated by mechanical simulation with random perturbation previously [Fig. 3D and Fig. 5C of https://doi.org/10.1242/dev.154609; Fig. 3 of https://doi.org/10.1088/1478-3975/ab6356]. Also, the diversified phenotypes within wild-type nematode embryos [Fig. 7 of https://doi.org/10.1242/dev.200401] or RNAi-treated nematode embryos [Fig. S10 of https://doi.org/10.1371/journal.pcbi.1009755] are not surprising, where experimental scientists usually only consider the highly reproducible phenotypes as the unreproducible ones can be from all kinds of unknown reasons.

12. Page 35, Line 708, “Biophysical parameters such as cell volume affect cell cycle duration…” —— There are two other pieces of literature that quantitatively measure how cell volume affects cell cycle duration during C. elegans embryogenesis [https://doi.org/10.1088/1367-2630/aaea91;
https://doi.org/10.1103/PhysRevE.104.054409].

Reviewer #3: This manuscript proposes pairwise metrics for characterizing differences between cell lineage trees and subtrees. The metrics proposed are the tree edit distance, which describes the structural difference between two trees, and the branch distance, which measures the difference between distributions of one phenotype on two trees. Branch distances are divided up into intersection and union distances where the intersection distance is restricted to parts of two trees that share the same topology.

The metrics are used to describe the variability of cell cycle times in wild type and RNAi-treated C. elegans lineage trees. Notch signalling is manifested in the shortening of cell cycle times and a distinctive pattern of branch distances between certain lineages, a pattern which is disrupted in embryos treated with RNAi against glp-1.

The authors also develop a heuristic to describe the proximity of cell cycle times between a homeotically-transformed lineage and wild-type lineages. Here samples of wild-type lineages are used to describe the neighborhood of the destination lineage to account for variability between wild-type embryos. They show that for a few genes, RNAi treatment results in cell cycle times in the transformed lineage match those in the destination lineage. Their results quantify the considerable variability in penetrance and severity for these genes.

This paper makes the argument that developing a set of measures for describing the distribution of phenotypes on cell lineage trees will provide quantitative insights into the mechanisms of development, and in fact other cellular processes. I strongly agree. I do have some minor questions and suggestions to improve clarity:

p.4 line 86: “the distinct problem in comparing phenotypic measurements aligned to lineages has been less extensively explored.” Can you give any references?

p.7 line 156: In the union branch distance, a 0 is imputed for cell positions which are not shared between the two trees. Can you justify this?

p.10 Fig 2: Can you show the line y=x in Figs 2A and 2D? This would be a helpful guide to the eye for the discussion about slopes on p.12. (An even more trivial suggestion is to swap the positions of Figs 2C and 2D to align with the arrangement in Fig 2A and 2B.)

p.11 Fig 3A: I am puzzled why there appears to be 21 embryos in the larger batch (and 9 in the smaller batch). Yet the discussion on p.20 consistently refers to the larger batch having 22 embryos.

p.13, Fig 4B: The color bars are too small.

p.14, Fig 4 caption: It is not clear that the 30 WT embryos belong to cluster 3. Can this be stated explicitly in the caption or main text? I understand this is made clear in S1 and the S1 dataset. Also, it appears that embryos treated with a specific RNAi type are not necessarily grouped into the same cluster. Can you state this explicitly?

p.17, Figs A, C, D: Do the fine-grained striations in these figures represent different embryos? I assume the cells from each lineal position from each embryo are grouped together on the plot.

p.22, line 381: “In some cases, this value is 0 but overlaps the WT origin lineage neighborhood (green square) suggesting that the RNAi perturbed lineage maintained its original fate, at least in terms of cell cycle timing.” This seems to suggest that your transformation efficiency heuristic cannot distinguish between unperturbed lineages and perturbed lineages since they both have a transformation efficiency of 0?

p.22: If I understand this argument correctly, you are only focused on the destination lineage as defined by the fate markers from Du et al. If the cycle time vector happens to be in the neighborhood of another lineage that is not the destination lineage, you would still say that it has no neighbors?

p.22, line 398: Is a transformation score the number of neighbours up to a maximum of 22? And the transformation efficiency is the score divided by 22?

p.23, line 415: You might clarify the sentence “(Fig 6B, 6C) noting a region of high density around WT neighborhoods (S7 Fig).”

p.24, line 432: Can you be more explicit about how your transformation efficiency/score relates to penetrance and phenotypic severity?

p.24, line 433: When you say penetrance and severity “are separable phenomena” do you mean “can be distinguished”?

p.25, line 467: GITHUB reference has no address or link.

p.31, line 607: Can you be more explicit about what you are randomizing? Is it the cells between all positions and embryos or just between embryos but at the same lineal position?

p.32, line 643: This was quite confusing. Can you state the null model that is being tested by this type of randomization?

p.35, line 693: Where were the “linear gradients of severity” described?

p.44: I found myself constantly flipping to the supporting figures. I would consider incorporating some of these into the main text.

S2: I found it very difficult to read the labels in this figure. Can the pixelation be made to have higher resolution?

S3: Should the y axis say “mean branch distance” instead of “mean branch edit distance”?

**Have the authors made all data and (if applicable) computational code underlying the findings in their manuscript fully available?**

Reviewer #1: Yes

Reviewer #2: Yes

Reviewer #3: Yes

PLOS authors have the option to publish the peer review history of their article (what does this mean?). If published, this will include your full peer review and any attached files.

Reviewer #1: No

Reviewer #2: No

Reviewer #3: No
---

## [Decision Letter · Decision Letter 1]

4 Dec 2023

Dear %TITLE% Shah,

We are pleased to inform you that your manuscript 'Novel metrics reveal new structure and unappreciated heterogeneity in Caenorhabditis elegans development' has been provisionally accepted for publication in PLOS Computational Biology.

Best regards,

Shaun Mahony

Academic Editor

PLOS Computational Biology

Pedro Mendes

Section Editor

PLOS Computational Biology

Reviewer's Responses to Questions

**Comments to the Authors:**

Reviewer #1: The authors have satisfactorily replied to my main criticism.

Reviewer #2: The authors addressed all of my comments.

Reviewer #3: I thank the authors for their responses to my questions. I have no further concerns, other than the following typos:

1) p.13, line 342 - wild type type

2) p.17, line 454 - is there a typo in the sentence "noting around WT neighborhoods than regions further away"?

3) p.18, line 476 - we use penetrance to refers (should be refer)

**Have the authors made all data and (if applicable) computational code underlying the findings in their manuscript fully available?**

Reviewer #1: Yes

Reviewer #2: Yes

Reviewer #3: Yes

PLOS authors have the option to publish the peer review history of their article (what does this mean?). If published, this will include your full peer review and any attached files.

Reviewer #1: **Yes: **Pablo Meyer

Reviewer #2: No

Reviewer #3: No

---

## [Editor Report · Acceptance letter]

13 Dec 2023

PCOMPBIOL-D-23-00893R1 

Novel metrics reveal new structure and unappreciated heterogeneity in Caenorhabditis elegans development

Dear Dr Shah,

I am pleased to inform you that your manuscript has been formally accepted for publication in PLOS Computational Biology. Your manuscript is now with our production department and you will be notified of the publication date in due course.

With kind regards,

Anita Estes
